

# Assessing the impact of meteorological forcing and its uncertainty on snow modeling and reanalysis

Haorui Sun[1], Steven A. Margulis[1]

[1]Department of Civil and Environmental Engineering, University of California, Los Angeles, Los Angeles, CA 90095, USA

*Correspondence to*: Steven A. Margulis (margulis@seas.ucla.edu)

**Abstract.** Large uncertainties in global model-based snow datasets, particularly in snow water equivalent (SWE), limit our understanding of snow storage and its response to climate change. These uncertainties are sensitive to meteorological inputs used to force offline snow models. In this study, we assessed the impact of three meteorological forcing datasets (i.e., ERA5, MERRA-2, and NLDAS-2) on ensemble SWE estimates within a probabilistic snow modeling and reanalysis framework

across three snow-dominated mountainous watersheds in the western US. Prior (open-loop) SWE estimates show significant inter-dataset variability, primarily driven by differences in cumulative snowfall. SWE errors are dominated by bias and no single-forcing dataset consistently outperforms the others across all domains or elevations. To assess the value of using multiple products, we construct a multi-forcing ensemble using least-square-based weighting informed by prior performance. The multi-forcing ensemble reduces errors compared to individual forcings and improves prior SWE accuracy across all

regions. Assimilation of near-peak lidar-derived snow depth substantially corrects prior SWE errors, reducing the influence of forcing-driven biases accumulated during the snowfall season. As a result, random error is the dominant source of posterior error. Although assimilation narrows performance differences, the multi-forcing ensemble still yields slightly better overall accuracy and improved uncertainty characterization. This work demonstrates that integrating diverse meteorological forcings within a data assimilation framework improves SWE estimates (both model-based and reanalysis-based), especially where the

optimal forcing dataset is uncertain.

## 1 Introduction

Seasonal snowpack in mountainous regions plays a crucial role in the global energy and water cycle. Mountains serve as water towers by storing water in snowpack and releasing snowmelt runoff during the warm season to supply downstream water demands and ecosystems. Such water towers are vulnerable to climatic and socio-economic changes, which can have negative

impacts on ~2 billion people (22% of global population) living downstream (Immerzeel et al., 2020; Mankin et al., 2015). Exceptional snow droughts can cause economic loss of billions of dollars, food shortages, and regional conflicts. Recent studies have documented substantial declines in snowpack across the western US (WUS), with over 90% of long-term monitoring sites exhibiting decreasing trends in April 1 SWE, including a 15-30% decline since the mid-20[th] century – equivalent to the volume of Lake Mead (Mote et al., 2018). Projections further suggest a 34 ± 8% loss in volumetric snowfall across the WUS



by the end of the century under high-emissions scenarios (Norris et al., 2025). Efforts to quantify snow droughts and their socio-economic impact have expanded across High Mountain Asia (HMA), Central Asia, western Russia, Andes, and Patagonia (Huning and AghaKouchak, 2020; Qin et al., 2020). Beyond snow droughts, recent studies have identified rain-on-snow (ROS) events as key drivers of major floods across North America (Li et al., 2019), including in mountainous regions like Alberta, where such events contributed to the 2013 flood (Mohammed et al., 2025; Pomeroy et al., 2016). Heavy rainfall

on top of snowpack increases runoff and poses a significant flood hazard. Snowpack should be characterized not only as a water supply but also as a potential source of hydrologic hazard.

Despite its importance, large uncertainties exist in the climatology of seasonal SWE magnitude and timing across different global datasets (e.g., ERA5, ERA5-Land, MERRA2, GLDAS, JRA55, GlobSnow; Mortimer et al., 2020). For example, Wrzesien et al. (2019) found that global datasets (ERA-Interim, GLDAS, MERRA2, VIC outputs) may underestimate snow

water storage by as much as 1500 km$^3$, which is equivalent to 4% of the rivers over the globe. Recent work further quantified climatological peak snow storage uncertainties, estimating $161 \pm 102$ km$^3$ over HMA (Liu et al., 2022), $284 \pm 14$ km$^3$ among high- and moderate-resolution products and $127 \pm 54$ km$^3$ among low-resolution products over the WUS, and $19 \pm 16$ km$^3$ across the Andes (Fang et al., 2023). In addition to these substantial uncertainties in bulk storage estimates, the spatial distribution of snow storage also varies across datasets and is resolution dependent. Coarse resolution products often fail to

capture snow storage patterns in transboundary mountain regions (e.g., Sierra Nevada and Andes) where snowmelt feeds watersheds that supply distinct downstream populations (Fang et al., 2023). These uncertainties in global snow datasets directly impact estimates of runoff, thereby leaving our physical understanding of mountain snowmelt driven systems still incomplete and highly uncertain.

The vast majority of gridded SWE datasets are model-derived, with limited observational constraints, making SWE

estimates highly sensitive to the meteorological inputs used as model forcing. These inputs govern both snowfall accumulation and snowmelt processes, with snowfall primarily controlled by precipitation and air temperature (Guan et al., 2010) and snow ablation influenced by solar radiation, snow albedo, air temperature, and atmospheric humidity (Cazorzi and Dalla Fontana, 1996; Harpold and Brooks, 2018). With the growing number of gridded meteorological datasets, the selection of snow model input data leads to substantial variability in snow simulations (Mizukami et al., 2014). Previous studies have evaluated different

gridded meteorological datasets in the context of hydrological modeling (Eldardiry et al., 2025; Kim et al., 2021; Raimonet et al., 2017; Wang et al., 2020; Wayand et al., 2013; Yoon et al., 2019). For example, Yoon et al. (2019) compared ten precipitation products over HMA and found that although the general spatiotemporal patterns are similar, significant differences in the mean estimates are observed across products and are propagated to the terrestrial water budget estimates. Models used to estimate SWE typically fall into two categories: coupled land-atmosphere models and offline land surface

models (LSMs). Coupled models, such as Earth System Models (ESMs) can be used for applications ranging from seasonal-scale prediction systems (e.g., NASA's GMAO S2S system; (Molod et al., 2020) to decadal and century-scale climate projections (e.g., NASA's GISS ModelE; (Kelley et al., 2020). A common feature of these models is that they generally produce estimates that are either significantly coarser than moderately sized watersheds or do not sufficiently capture the



spatial variability within large watersheds. They also frequently exhibit significant biases (Duethmann et al., 2013; Emmanouil et al., 2021; Mamalakis et al., 2017; Seyyedi et al., 2014), which along with their coarse spatial resolution, limit their utility for hydrologic prediction in moderately sized to large watersheds without additional bias correction and downscaling. Offline LSMs are often used to reduce the influence of these biases by using bias-corrected meteorological inputs. However, SWE estimates in both frameworks remain sensitive to the characteristics and quality of the input data used.

Data assimilation (DA) techniques aim to constrain SWE estimates by optimally merging model simulations with observational data such as snow depth and snow cover area, thereby correcting prior model outputs (Andreadis and Lettenmaier, 2006; Fang et al., 2022b; Liu et al., 2021; Margulis et al., 2015; Slater and Clark, 2006; Smyth et al., 2019, 2020). In this study, we adopt an offline LSM to generate a prior ensemble, which represents a range of possible snow states before assimilation. These prior estimates are driven by meteorological forcings with modelled uncertainties in key variables such as precipitation, air temperature, and radiation. The prior estimates are highly sensitive to the meteorological forcings, especially in complex mountainous regions with limited in-situ observations (Raleigh et al., 2015; Slater et al., 2013). Raleigh et al. (2015) demonstrated through global sensitivity analyses that different choices of perturbation functions and error distributions can substantially influence prior model outputs. Although DA frameworks are designed to ultimately reduce these a priori uncertainties, the effectiveness is limited by how well meteorological forcing uncertainty is represented (Girotto et al., 2020). Most existing DA studies rely on a single meteorological dataset (e.g., interpolated weather station data or gridded reanalysis datasets) to generate a prior ensemble, often without explicitly accounting for inconsistencies across different datasets. Clark et al. (2011) advocate for the multiple working hypotheses for hydrological modeling, emphasizing the need to test alternative model structures rather than assuming a single "correct" representation. Similarly, assuming a single "correct" forcing dataset with perturbations can limit our understanding of uncertainty propagation in SWE estimates. An ensemble-based data assimilation framework fits naturally within the multiple hypotheses approach, allowing us to examine how differences in forcing datasets, error characteristics, and uncertainty perturbations influence SWE estimates.

The objective of this study is to examine how differences among meteorological datasets influence both model-based (prior) and DA-based (posterior) SWE estimates, and to assess whether using multiple datasets can improve performance relative to using any single dataset. We implement an ensemble Bayesian snow reanalysis framework that integrates multiple meteorological datasets, explicitly accounts for their uncertainties, and conditions prior snow depth estimates on lidar-based snow depth observations as a case study. This framework allows us to address the following questions:

(1) Does one of the readily available meteorological forcing datasets yield the most accurate model-based prior SWE spatio-temporal estimates?

(2) To what extent does using multiple meteorological forcing datasets improve the accuracy of model-based prior SWE estimates compared to any single forcing?

(3) To what extent can assimilation of independent data reduce errors arising from meteorological forcing in model-based prior SWE estimates?



(4) How does the incorporation of multiple meteorological forcing datasets influence the accuracy of DA-based posterior SWE estimates and their uncertainty?

The rest of the paper is organized as follows: Section 2 presents the case study domains and methodology used in this work. Section 3 provides the results and discussion to answer the questions listed above, and Section 4 summarizes the key points of this work.

## 2 Methods

### 2.1 Case study domains and water year testbed

The case study domains include three mountainous watersheds in the WUS: the Merced River Basin, Aspen-Castle Maroon (hereafter Aspen), and Gunnison-East. The locations of the domains are shown in Fig.1(a). These watersheds are selected because (1) they are representative of seasonally snow-dominated WUS sites and provide important water resources through spring snowmelt, (2) the snow reanalysis method employed herein has been successfully applied over these sites previously (Fang et al., 2022), and (3) lidar-based snow depth data near April 1st is available as a constraint for data

assimilation, and additional lidar-based observations later in the snowmelt season are available for evaluation during the melt season. Water year (WY) 2019 is selected for analysis due to the availability of lidar-based observations across all three domains at both near-peak SWE and in the melt season.

    The Merced River Basin is in the western Sierra Nevada of California and corresponds to an Airborne Snow Observatory (ASO) surveyed area of approximately 966 km², with elevations ranging from about 1195 m to 3861 m. It

receives winter precipitation predominantly as snow at higher elevations, forming a seasonal snowpack that melts in spring and early summer. The snowmelt contributes to downstream river flow as it eventually drains westward into the San Joaquin River in California's Central Valley. The Aspen watershed is located within the Roaring Fork watershed in central Colorado on the western side of the Continental Divide. It spans an ASO surveyed area of about 369 km², with elevations ranging from 2532 m to 4284 m. Snowmelt from this watershed feeds Castle and Maroon Creeks, which largely flow northward and supply

almost all the water for the city of Aspen. Adjacent to Aspen, the Gunnison-East watershed, part of the Gunnison River Basin, covers an ASO surveyed area of 1342 km², with elevations ranging from 2529 m to 4172 m. It receives substantial winter precipitation as snow, with spring snowmelt feeding major tributaries that flow southeast into the Gunnison River and ultimately into the Colorado River.

### 2.2 Meteorological datasets for forcing a snow model

This study uses hourly meteorological forcings that include surface precipitation, 2-m air temperature and specific humidity, surface pressure, surface downwelling shortwave radiation, and 10-m wind speed. These variables are derived from three widely used datasets – ECMWF Reanalysis v5 (ERA5) (Hersbach et al., 2020), Modern-Era Retrospective analysis for



Research and Applications v2 (MERRA-2) (Gelaro et al., 2017), and North American Land Data Assimilation System Phase 2 (NLDAS-2) (Xia et al., 2012). These datasets were selected because they are large-scale, gridded hourly products developed by different agencies using distinct atmospheric models, assimilation systems, and observational constraints. They have been used in a variety of modeling and data assimilation applications (e.g., Lim et al., 2017; Margulis et al., 2016; Tao et al., 2019; Tarek et al., 2020). While additional forcing datasets are available, triplet configurations are commonly used to derive weights and characterize uncertainty (Yilmaz et al., 2012). ERA5 is a global dataset generated based on the Integrated Forecasting System and a 4D-Var data assimilation system that integrates satellite-based observations and station data (Hersbach et al., 2020). It provides hourly output at 0.25° × 0.25° resolution from 1979 onward. MERRA2 is developed by NASA's Global Modeling and Assimilation Office (GMAO). It uses the GEOS-5 atmospheric model and a 3D-Var assimilation system, incorporating satellite observations and conventional meteorological observations (Gelaro et al., 2017). It provides global data at 0.625° × 0.5° resolution from 1980 to present. In contrast, NLDAS2 is a Contiguous U.S. product that combines model output from the NCEP North American Regional Reanalysis (NARR) with high-resolution observational data. It replaces NARR precipitation with gauge-based estimates, utilizes GOES satellite-derived radiation, and adjusts near-surface variables using dense surface station networks (Xia et al., 2012). It provides data at a 0.125° × 0.125° resolution from 1979 to present.

Figure 1 (b) shows the raw annual surface precipitation for WY 2019 from ERA5, MERRA2, and NLDAS2 across the study domains, with domain-averaged mean and standard deviations annotated for each dataset. Differences in precipitation magnitude and spatial variability are significant due to variations in spatial resolution and estimation methods, which are expected to propagate to substantial uncertainties in hydrological model outputs (Tang et al., 2023). In Merced, ERA5 and MERRA2 exhibit similar mean precipitation values, with both substantially higher than NLDAS2. However, MERRA2 shows significant spatial contrast across the watershed due to the coarse resolution. In Aspen and Gunnison-East, the three products diverge more significantly: MERRA2 exhibits higher precipitation, while ERA5 and NLDAS2 are more similar to each other. To reduce the large spatial contrast seen in the coarse-gridded datasets, a bilinear interpolation step was applied to smooth the abrupt spatial artifacts prior to downscaling as described below in Sect. 2.3.



**Figure 1**. (a) Map of elevation and Hydrological Unit Codes 2 (HUC2) basins in the western US with the locations of study domains, (b) maps of the raw annual precipitation (mm) for the Water Year (WY) 2019 from ERA5, MERRA2, and NLDAS2 across the reanalysis tiles that cover the study domains (top row showing Merced and bottom row showing Aspen and Gunnison). The spatial mean and standard deviation (Std. Dev.) of precipitation for each domain are annotated on the maps.



## 2.3 Downscaling and ensemble uncertainty modeling of meteorological forcings

This study uses a modeling spatial resolution of 150 m, which represents a relatively high-resolution capable of resolving topographic variability relevant to mountain snow processes, while remaining computationally feasible for the study domains. However, as is commonly the case, the native resolutions of meteorological forcing datasets (ERA5, MERRA2, and NLDAS2) are much coarser than the model grid. Therefore, a downscaling approach is required to adjust the coarse inputs to the finer model resolution. To address this mismatch, we apply topographic corrections following the method of Girotto et al. (2014), as detailed in Sect. S1. This approach has been successfully applied in snow studies over various mountainous regions, including the WUS (Fang et al., 2022), HMA (Liu et al., 2021), and the Andes (Cortés and Margulis, 2017). Detailed procedures for downscaling air temperature, dew point temperature, surface pressure, specific humidity, and incoming shortwave and longwave radiation are provided in Sect. S1.

In addition to downscaling, bias correction is important to account for known systematic errors in the meteorological forcing datasets, particularly when generating prior ensembles for data assimilation. Previous studies have documented biases in downscaled MERRA2 (Fang et al., 2022; Liu et al., 2021; Cortés and Margulis, 2017) and NLDAS2 (Margulis et al., 2016) datasets, and similar biases are expected in ERA5. To address these first-order biases and enable a probabilistic modeling framework, we apply a uniform prior bias correction approach that adjusts each forcing dataset across the whole WUS. Following downscaling, key meteorological variables including precipitation, air temperature, dew point temperature, and shortwave radiation are bias-corrected and perturbed to generate prior ensemble realizations. The ensemble generation methodology follows that used in previous MERRA2-based reanalyses over the WUS (Fang et al., 2022) and HMA (Liu et al., 2021). Specifically, precipitation is adjusted using a lognormally distributed multiplicative factor (Eq. S1), while air temperature ($T_a$) and dew point temperature ($T_d$) are perturbed with normally distributed additive error (Eqs. S2 and S3). Shortwave radiation ($SW$) is corrected with a normally distributed multiplicative factor that varies with the solar index (Eq. S4). These first-order prior bias correction parameters are applied uniformly in space for each forcing dataset across the WUS.

In this study, snowfall model inputs are derived from the prior ensemble of bias-corrected precipitation and air temperature on an hourly basis (Sect. S1). Figure 2 shows the time series of basin-averaged daily cumulative prior mean snowfall, elevational binned histograms of annual snowfall, and spatial maps of annual snowfall for WY 2019 across the three study domains. The basin-averaged annual snowfall estimates vary across different datasets, ranging from 1553 mm to 2088 mm for Merced, from 798 mm to 1710 mm for Aspen, and from 694 mm to 1363 mm for Gunnison-East. Differences among datasets consistently exist across many elevation bands due to the coarse resolution of the raw forcing products and the use of the same deterministic downscaling approach. MERRA2 exhibits higher snowfall across all elevation bands, while ERA5 and NLDAS2 yield lower values, particularly at higher elevations. ERA5 exhibits the lowest snowfall except at low elevations in Gunnison-East. These differences persist despite the application of a uniform prior bias correction designed to reduce the first-order biases in the forcing datasets. Note that since the bias correction was derived using multiple sites across the WUS (Sect. S2), it is expected that the impact of that bias correction can vary significantly at more localized scales, as illustrated here for





the Merced, Aspen, and Gunnison-East basins. The observed inter-product variability reflects a combination of inherent differences in the raw datasets and the impacts of the applied bias corrections. These variations are expected to propagate into significant differences in SWE estimates, as discussed in Sect. 3.

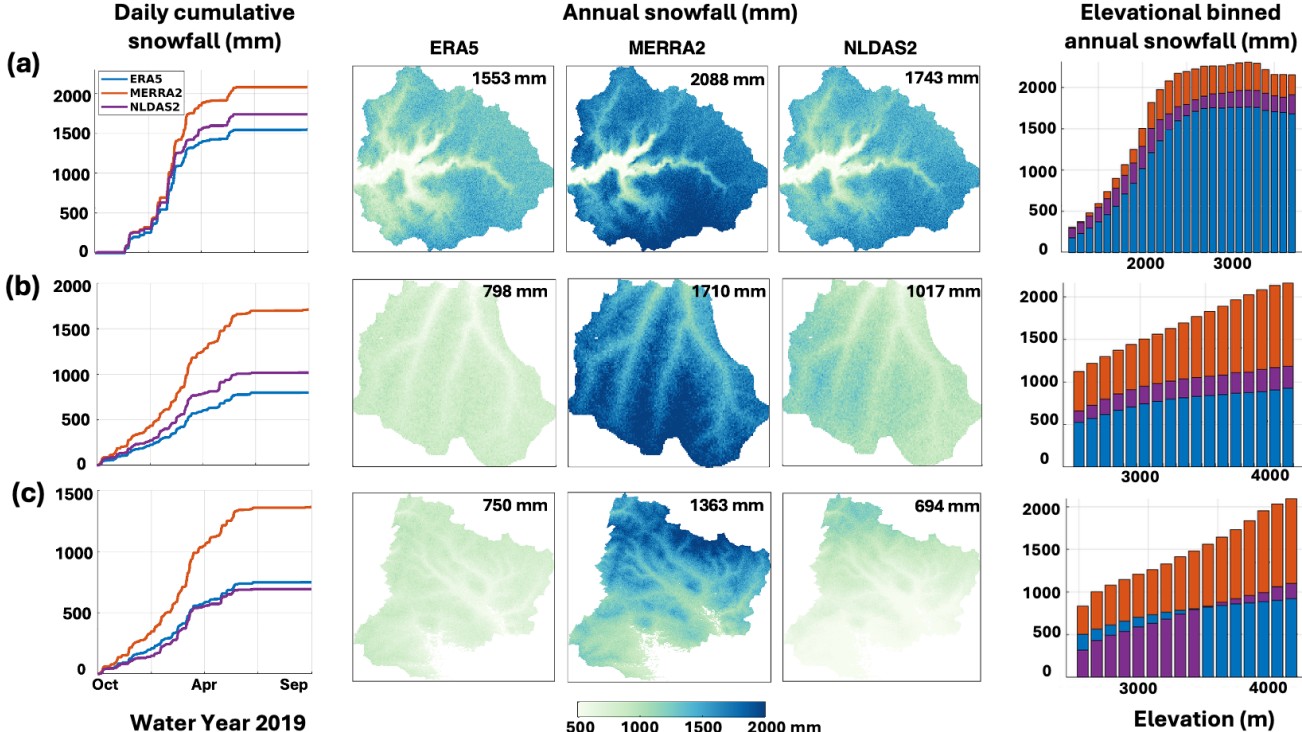

**Figure 2.** Time series of prior mean basin-averaged daily cumulative snowfall, spatial distribution of prior mean annual snowfall, and elevationally binned histograms of annual snowfall for Water Year (WY) 2019 over (a) Merced, (b) Aspen, and (c) Gunnison-East. Each row corresponds to a different basin. Basin-averaged annual snowfall values are annotated on the spatial maps.

## 2.4 Bayesian SWE estimation framework

### 2.4.1 Ensemble model framework

This study applies a previously developed ensemble modeling framework to explicitly represent uncertainties in meteorological forcings and evaluate their impact on SWE estimates (both model-based prior estimates and data-constrained posterior estimates). The modeling setup is the same as Fang et al. (2022). Specifically, the Simplified Simple Biosphere model coupled with a three-layer Snow Atmosphere-Soil Transfer model (SSiB3-SAST; Sun and Xue, 2001) was used as the land surface model (LSM) to simulate snow states. A snow depletion curve (SDC) was used to represent the subgrid heterogeneity





in snow states. Static inputs required by the LSM, including topographic, land cover, and forest cover fraction, are obtained
from the same sources and processed to the model resolution using the same methods as described in Fang et al. (2022).

Simulations are conducted at a spatial resolution of 150 m and an hourly time step for WY 2019, generating 120
ensemble members of snow estimates. This ensemble size was selected to ensure robust sampling of uncertainty in
meteorological forcings while maintaining computational efficiency. The ensemble members differ only in their
meteorological forcing inputs and bias correction parameters. The uncertainty model for each forcing dataset, described in
Sect 2.3, is used to generate a prior forcing ensemble to drive the LSM.

### 2.4.2 Assimilation of near-peak snow depth

The probabilistic DA framework used in this study is referred to as the Particle Batch Smoother (PBS) developed by
Margulis et al. (2015). Within this framework, the LSM is used to generate a prior ensemble estimate of snow states, based on
the specified input uncertainty and its propagation through the model. Each member in the prior ensemble is treated as an
equally likely realization. The goal of the PBS approach is to optimally weight uncertainties from prior estimates and snow
observations, in this case study, snow depth, to generate posterior estimates of snow states. A likelihood function is used to
update the weights of each prior ensemble member based on its consistency with the snow depth observation. The updated
(posterior) weights define the probability distribution of modelled snow variables (e.g., mean and variance). The mathematical
formulation is described in detail in Margulis et al. (2015, 2019).

In this study, a single snow depth (SD) map from the Airborne Snow Observatory (ASO, Painter et al., 2016),
collected near peak accumulation (closest to April 1$^{st}$), is assimilated to constrain the prior ensemble. This assimilation timing
is chosen to separate, to some extent, the effects of meteorological forcing errors during the accumulation season from those
during the melt season. Snow depth is tightly coupled to instantaneous SWE (Margulis et al., 2019), and near-peak SD
observations are expected to primarily correct errors during the snow accumulation season (i.e. primarily due to snowfall
inputs). This enables a more accurate initialization of the snowpack antecedent to melt and allows us to evaluate how
differences in meteorological forcing inputs influence differences in SWE evolution during the melt season. Before
assimilation, the ASO snow depth maps were regridded to the modelling resolution at 150 m. The snow depth observation is
assumed to be unbiased with a specified error standard deviation of 10 cm, which is in the correct range of lidar snow depth
errors (Painter et al., 2016) and the modeling resolution of this study. While lidar-derived SD data are not widely available,
they provide an ideal case study for evaluating the value of assimilating independent, high-quality snow observations. Given
the direct connection between SD and SWE, the case study with assimilation of SD is likely to provide a lower bound on
posterior SWE error, that may be larger when assimilating other variables (e.g. fractional snow-covered area).

### 2.4.3 Experimental setup

To assess the impact of meteorological forcings on SWE estimates, two sets of experiments were designed: single-
forcing experiments (1a and 2a) and multi-forcing experiments (1b and 2b), as defined in Table 1 and described below. These





experiments are grounded in an ensemble modelling and DA framework, where each ensemble member represents a plausible realization of the forcing time series. Typically, ensemble members are generated by perturbing a single nominal forcing dataset; however, in this study, we also explore an alternative approach where different forcing datasets are used to contribute realizations to the ensemble.

240

**Table 1.** Summary of Experimental Design for Evaluating SWE Estimates

| Experiment | Forcing Type | Assimilation | Experimental Goal |
|---|---|---|---|
| 1a | Single-forcing | No (model-based Prior) | Establish baseline SWE uncertainty when using a single meteorological forcing dataset. |
| 1b | Multi-forcing | No (model-based Prior) | Assess if combining multiple forcing datasets can reduce uncertainty compared to the single-forcing baseline. |
| 2a | Single-forcing | Yes (DA-based Posterior) | Quantify how much data assimilation improves SWE estimates within a single-forcing framework. |
| 2b | Multi-forcing | Yes (DA-based Posterior) | Evaluate if a multi-forcing ensemble leads to greater improvements in posterior accuracy and reduction in uncertainty compared to a single-forcing ensemble. |

## a. Single-forcing experiments

The single-forcing experiments (1a and 2a) are designed to assess how SWE estimates differ when using different
245  forcing datasets, while keeping all other model components fixed. The setup is the standard approach commonly used in snow modeling and data assimilation, where a single reanalysis or gridded dataset is selected as the nominal driver for the modeling system. Each 120-member ensemble is generated using bias-corrected and perturbed meteorological inputs from one dataset (ERA5, MERRA2, or NLDAS2). The output SWE provides a reference for comparison against the multi-forcing ensemble. Therefore, the single-forcing experiment serves as a baseline in addressing all four research questions, including the relative
250  accuracy of different forcing products, the value of using multiple products, and the value of data assimilation in reducing forcing-related SWE uncertainty.

## b. Multi-forcing experiments

Rather than merging meteorological inputs deterministically, the multi-forcing experiments leverage the ensemble-based framework to sample from the realizations associated with each individual forcing dataset. This approach is expected to
255  generate a more diverse set of meteorological inputs, as it draws from multiple nominal products, each with its own variability. The 120-member prior ensemble is populated using subsets of realizations drawn directly from the single-forcing ensembles. Each realization uses meteorological inputs from only one of the three forcing datasets to ensure internal physical consistency within each meteorological ensemble member.

A central design decision involves determining how many realizations to allocate to each product within the 120-
260  member ensemble. This is addressed by assigning partitioning weights, $W_i$, that specify the proportion of the ensemble



attributed to each dataset. One defensible approach is to use weights that are calculated using the least squares weighting approach developed by Yilmaz et al. (2012), which derives optimal weights based on the relative uncertainties of each product under the assumption of additive, Gaussian, and uncorrelated errors. The partitioning weights are calculated as:

$$W_1 = \frac{\sigma_{\epsilon_2}^2 \sigma_{\epsilon_3}^2}{\sigma_{\epsilon_1}^2 \sigma_{\epsilon_2}^2 + \sigma_{\epsilon_1}^2 \sigma_{\epsilon_3}^2 + \sigma_{\epsilon_2}^2 \sigma_{\epsilon_3}^2} \qquad \text{Eq. 1}$$

$$W_2 = \frac{\sigma_{\epsilon_1}^2 \sigma_{\epsilon_3}^2}{\sigma_{\epsilon_1}^2 \sigma_{\epsilon_2}^2 + \sigma_{\epsilon_1}^2 \sigma_{\epsilon_3}^2 + \sigma_{\epsilon_2}^2 \sigma_{\epsilon_3}^2} \qquad \text{Eq. 2}$$

$$W_3 = \frac{\sigma_{\epsilon_1}^2 \sigma_{\epsilon_2}^2}{\sigma_{\epsilon_1}^2 \sigma_{\epsilon_2}^2 + \sigma_{\epsilon_1}^2 \sigma_{\epsilon_3}^2 + \sigma_{\epsilon_2}^2 \sigma_{\epsilon_3}^2} \qquad \text{Eq. 3}$$

where $W_i$ is the weight assigned to forcing product i and $\sigma_{\epsilon_i}^2$ is the estimated error variance in modelled SWE relative to ASO-derived SWE at the time of assimilation (near April 1st). These error variances are approximated using the root-mean-square error (i.e., $\sigma_{\epsilon_i}^2 = \text{RMSE}_i{}^2$), which incorporates both systematic and random errors. RMSE can be decomposed into bias and unbiased RMSE (ubRMSE) components as $\text{RMSE}^2 = \text{bias}^2 + \text{ubRMSE}^2$ (Entekhabi et al., 2010). While the original formulation (Yilmaz et al., 2012) assumes unbiased inputs such that RMSE=ubRMSE, we relax that assumption here by using independent data and taking the full RMSE to represent total error. Since this weighting approach requires a reference SWE dataset, it is implemented herein as a case study at sites where ASO data are available. Sensitivity of the results to the weighting scheme is further evaluated in Sect. 3.6. This weighting method constructs the multi-forcing ensemble as a weighted combination of the individual single-forcing ensembles. If one dataset performs significantly better than the others, the weighted average may underperform relative to that best case. However, in most applications, it is unlikely to know which forcing product is most accurate in advance, nor can one assume a single product is consistently best across space and time. Given this uncertainty, the goal is not necessarily to outperform the best product, but to avoid over-reliance on a single dataset and to reduce errors in cases where no single product consistently dominates.

The number of realizations derived from each dataset is then given by:

$$N_1 = W_1 \times N \qquad \text{Eq. 4}$$

$$N_2 = W_2 \times N \qquad \text{Eq. 5}$$

$$N_3 = W_3 \times N \qquad \text{Eq. 6}$$

where $W_i$ denotes the partitioning weights (Eqs. 1-3) for ERA5, MERRA2, and NLDAS2 respectively. Note that the sum of $W_i$ is equal to 1. This setup ensures that $N_1 + N_2 + N_3 = N$, so that the total number of realizations in the multi-forcing ensemble (i.e., 120) is the same as that of the single-forcing experiment, allowing for a fair comparison under equal computational cost.

Figure 3 illustrates the Bayesian reanalysis framework, with the forward model (red boxes) and lidar-based ASO snow depth assimilation (blue boxes). The "Multi-forcing" block shows how individual realizations are sampled from ERA5, MERRA2, and NLDAS2 single-forcing ensembles, with partitioning weights applied to determine their contributions.



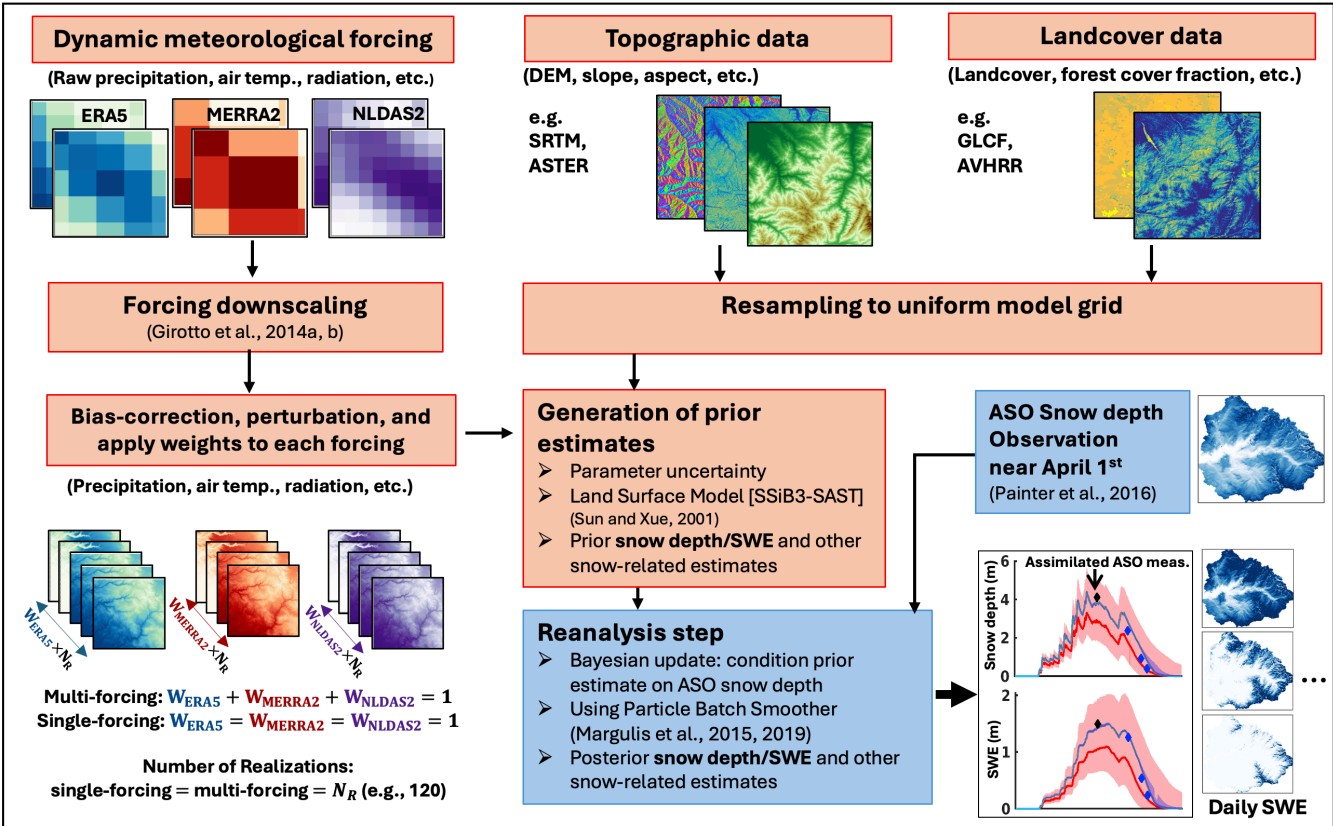

**Figure 3.** Flow chart of the Bayesian snow reanalysis framework, which consists of an ensemble-based prior modeling system (red boxes) and a posterior update component for assimilating ASO snow depth (SD) near April 1st (blue boxes). In the single-forcing experiments, prior ensemble members correspond to single forcing product $i$ with a weight $W_i = 1$, whereas the multi-forcing experiments use multiple products with weights that sum to 1.0.

**2.5 SWE reference dataset**

The performance of both single-forcing and multi-forcing SWE estimates is evaluated through comparisons with spatially distributed SWE derived from ASO SD observations. Note that the ASO SWE product is not directly used for the comparison since it is a model-based estimate that combines model-estimated snow density and SD observations. Raleigh and Small, (2017) found that snowpack density is the primary source of uncertainty when mapping basin-wide SWE with lidar. Painter et al. (2016) pointed out that ASO snow density is estimated using a spatially distributed energy-balance model, iSnobal (Marks, 1999), which represents the snow cover as a two-layer system. However, there are inconsistencies in the snow density estimates compared to those from the SSiB3-SAST model used in this study and it is difficult to know which density estimates are more accurate. Analyzing how model-based snow density uncertainties affect the SWE estimates is beyond of the scope of this study. To reduce the influence of model-dependent snow density differences in the comparison, we instead compute an





"ASO-based SWE" reference product used herein by multiplying observed ASO SD with the mapped ensemble mean snow density estimates across the three single-forcing ensembles at each of the ASO observation times. This approach provides a more internally consistent comparison. Analysis of estimated snow density at ASO times indicates limited variability across the three single-forcing datasets (Sect. S3).

## 3 Results and Discussion

### 3.1 Experiment 1a: Impact of single dataset meteorological forcings on model-based (prior) SWE

Daily time series of basin-averaged SWE and snow depth (SD) for WY 2019 (Fig. 4) clearly show the substantial differences in the prior mean estimates derived from the three meteorological forcing datasets. MERRA2 consistently produces the highest SWE and SD across all three domains, showing positive biases relative to the ASO-based reference. In contrast, ERA5 and NLDAS2 typically yield lower SWE and SD estimates, which generally better align with the ASO-based SWE, except in Gunnison-East, where they exhibit significant negative differences relative to the ASO-based reference in April. No single dataset consistently outperforms the others across all regions, and this is likely true at larger spatial scales. Based on peak SWE near April 1$^{st}$ (coinciding with ASO acquisition), the relative agreement with ASO varies by basin. For Merced, ERA5 is the closest to the ASO-based reference, followed by NLDAS2 and MERRA2. For Aspen, NLDAS2 performs best, followed by ERA5 and MERRA2. For Gunnison-East, MERRA2 aligns most closely with ASO, followed by ERA5 and NLDAS2. On the near-peak ASO date, the range of SWE variations across the three datasets are 0.46 m in Merced, 0.64 m in Aspen, and 0.51 m in Gunnison-East. Corresponding SD differences are 1.03 m, 1.69 m, and 1.29 m, respectively. These differences in SWE and SD estimates can be partially explained by the snowfall patterns shown in Fig. 2. MERRA2 exhibits the highest annual snowfall, particularly at higher elevations (Fig. 2), propagating to its overestimation of peak SWE relative to the other datasets. The overestimation of peak SWE generally propagates to the overestimation of SWE during the ablation season, since ablation errors are due to both peak SWE errors and forcing errors in simulating snowmelt. Constraining peak SWE in the DA step isolates ablation errors primarily due to meteorological forcing, as discussed in Sect. 3.5 and 3.6.





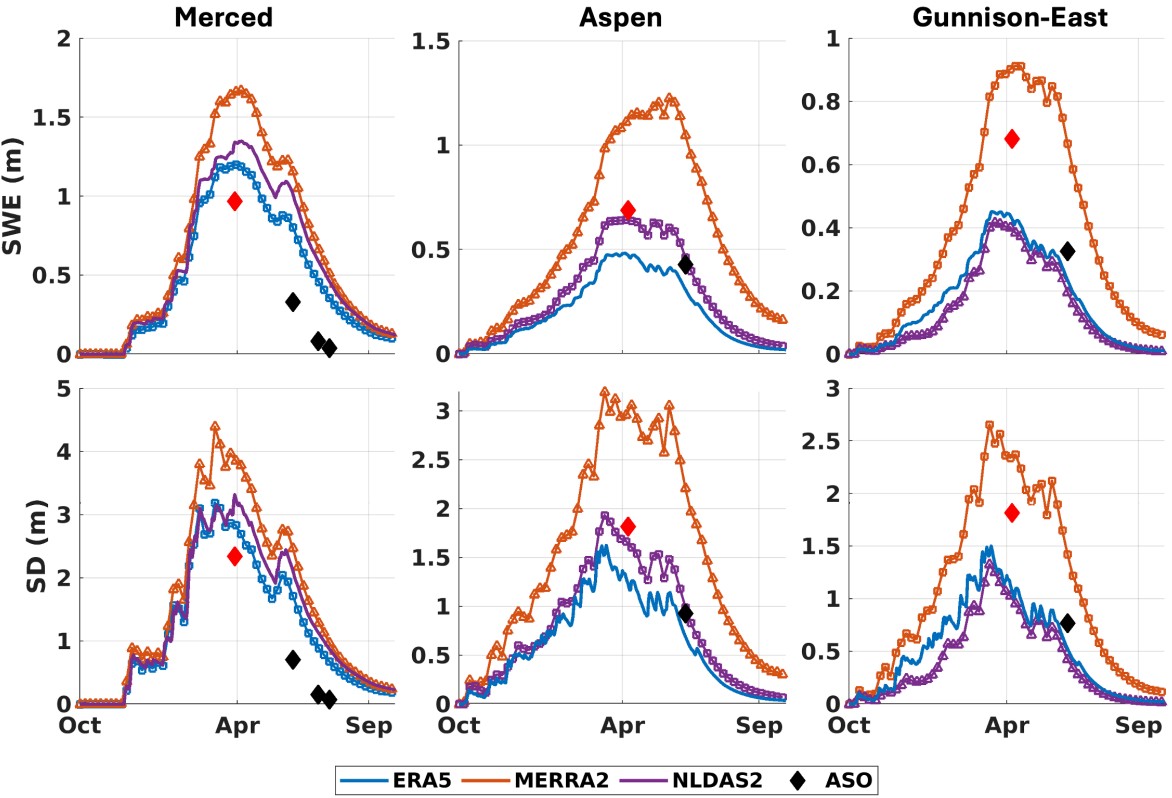

**Figure 4.** Daily time series of basin-averaged prior ensemble mean SWE and SD estimated using ERA5, MERRA2, and NLDAS2 forcings for Water Year 2019 across Merced, Aspen, and Gunnison-East. ASO SD and ASO-based SWE estimates are shown as diamonds, with red diamonds indicating values on the near-peak ASO date. The best- and worst-performing dataset on the near-peak ASO date, based on comparison with ASO, are highlighted with squares and triangles, respectively.

Figure 5 illustrates the elevational distribution of prior mean SWE produced using the three forcing datasets, compared to ASO-based SWE. All three prior estimates exhibit relatively simplistic elevational distributions that primarily reflect cumulative snowfall differences (Fig. 2), which project to shifts in SWE across elevation. MERRA2 consistently yields higher SWE across all elevation bands compared to both the ASO and other single-forcing prior estimates. While ERA5 and NLDAS2 produce lower SWE than MERRA2 across all domains, their differences relative to ASO vary by region and elevation. Note that ASO shows a more complex variability pattern that none of the prior estimates capture particularly well, even when their average SWE magnitudes are similar. In Merced, both ERA5 and NLDAS2 overestimate SWE relative to ASO, and their SWE estimates indicate later snowmelt, particularly at lower elevations. This delay is likely due to the positive bias in peak SWE, as even accurate melt rates require longer time to melt out more snowpack. In Aspen, the basin-averaged agreement of NLDAS2 with ASO (Fig. 4) results from compensating differences across elevations – overestimating SWE at low elevations and underestimating at high elevations. ERA5 most closely matches ASO below 3200 m, while NLDAS2 is



better aligned between 3200 m and 3700 m. In Gunnison-East, ERA5 aligns best with ASO below 3200 m, whereas MERRA2 provides the closest match between 3200-3800 m. The ranking of prior SWE accuracy is a function of elevation.

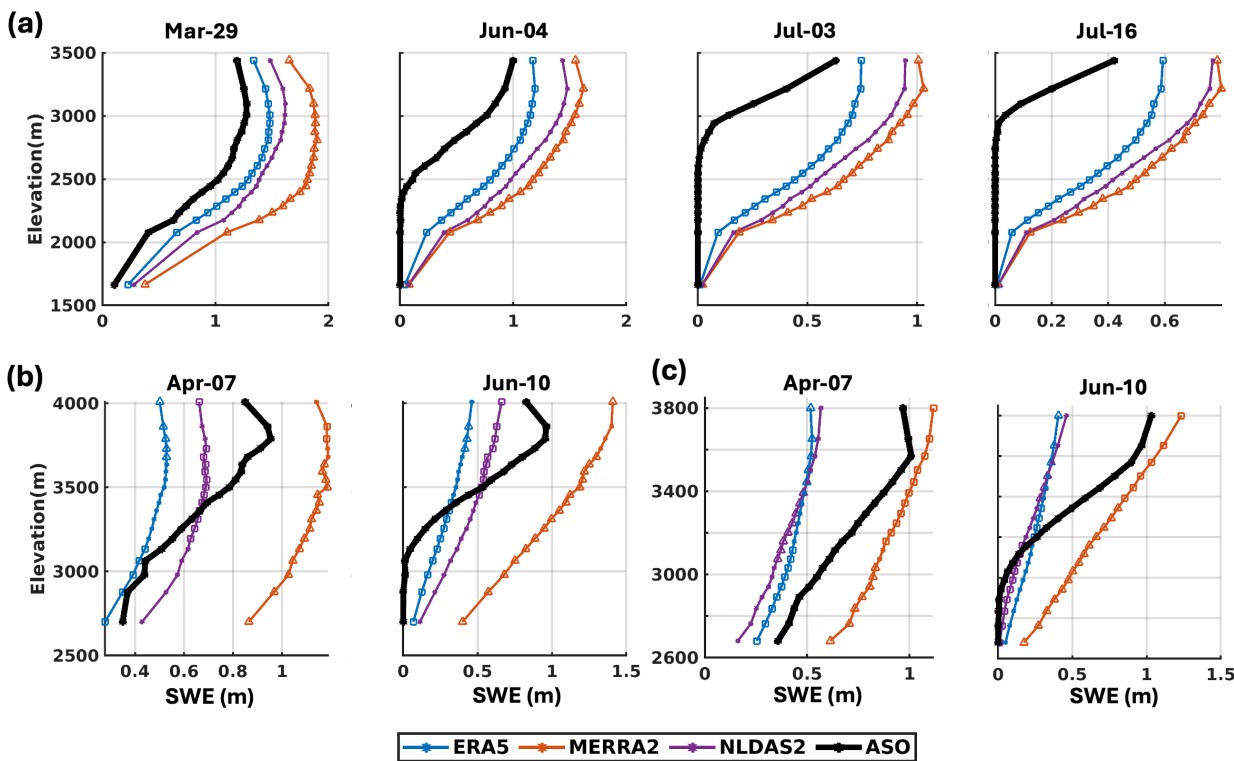

**Figure 5.** Elevational distribution of ASO-based SWE and prior mean SWE produced using ERA5, MERRA2, and NLDAS2
forcings across (a) Merced, (b) Aspen, and (c) Gunnison-East. Elevation is discretized into 20 bins containing equal numbers of pixels, with mean SWE values within each bin shown as solid curves. Each elevation bin contains an equal number of pixels. Mean SWE values within each elevation bin are shown as solid curves. Datasets that best and worst match the ASO data are indicated by squares and triangles, respectively.

The accuracy of prior mean SWE is assessed using RMSE, calculated relative to ASO-based SWE (Table 2). In Merced, the RMSE for ERA5 ranges from 0.36 m to 0.58 m, for MERRA2 from 0.54 m to 0.92 m, and for NLDAS2 from 0.48 m to 0.78 m throughout the year. ERA5 and NLDAS2 exhibit similar RMSE in the range of 0.3-0.45 m in Aspen and Gunnison-East. Figure 6 presents the spatial patterns of SWE differences between the prior mean estimates and ASO-based estimates, along with a decomposition of RMSE (into its biased and unbiased components). The top panel illustrates that
MERRA2 consistently overestimates SWE across most regions in all domains (indicated by widespread blue areas), whereas ERA5 and NLDAS2 exhibit fewer positive differences in Merced, localized underestimations in Aspen, and more negative differences in Gunnison-East. The bottom panel shows the RMSE of prior mean SWE which is chosen since it provides an




easy way to decompose total error into bias and ubRMSE components. MERRA2 exhibits the largest RMSE in Merced and Aspen, primarily due to large bias contributions (with $\text{bias}^2/\text{RMSE}^2$ ratio exceeding 0.5), while ERA5 and NLDAS2 show

smaller and less biased errors, particularly in Aspen. In contrast, MERRA2 exhibits lower RMSE in Gunnison-East on April 7th, 2019, where random errors dominate. For ERA5 and NLDAS2, ubRMSE is the main contributor to RMSE during the snowmelt period (i.e., June 10th, 2019) in Aspen and Gunnison-East. Overall, differences in the bias are mostly larger than those in ubRMSE across the three datasets. The dataset ranking based on the RMSE of prior mean SWE at the time of the ASO observation near April 1st is consistent with the ranking of the basin-averaged SWE errors (Fig. 4).


**Table 2.** RMSE (m) of prior mean SWE, compared to ASO-based SWE for Water Year 2019.

| Domain | Date | ERA5 | MERRA2 | NLDAS2 |
|---|---|---|---|---|
| Merced | Mar-29 | 0.36 | 0.77 | 0.48 |
| | Jun-04 | 0.58 | 0.92 | 0.78 |
| | Jul-03 | 0.47 | 0.68 | 0.59 |
| | Jul-16 | 0.39 | 0.54 | 0.49 |
| Aspen | Apr-07 | 0.37 | 0.53 | 0.31 |
| | Jun-10 | 0.43 | 0.72 | 0.40 |
| Gunnison-East | Apr-07 | 0.39 | 0.36 | 0.41 |
| | Jun-10 | 0.33 | 0.49 | 0.34 |



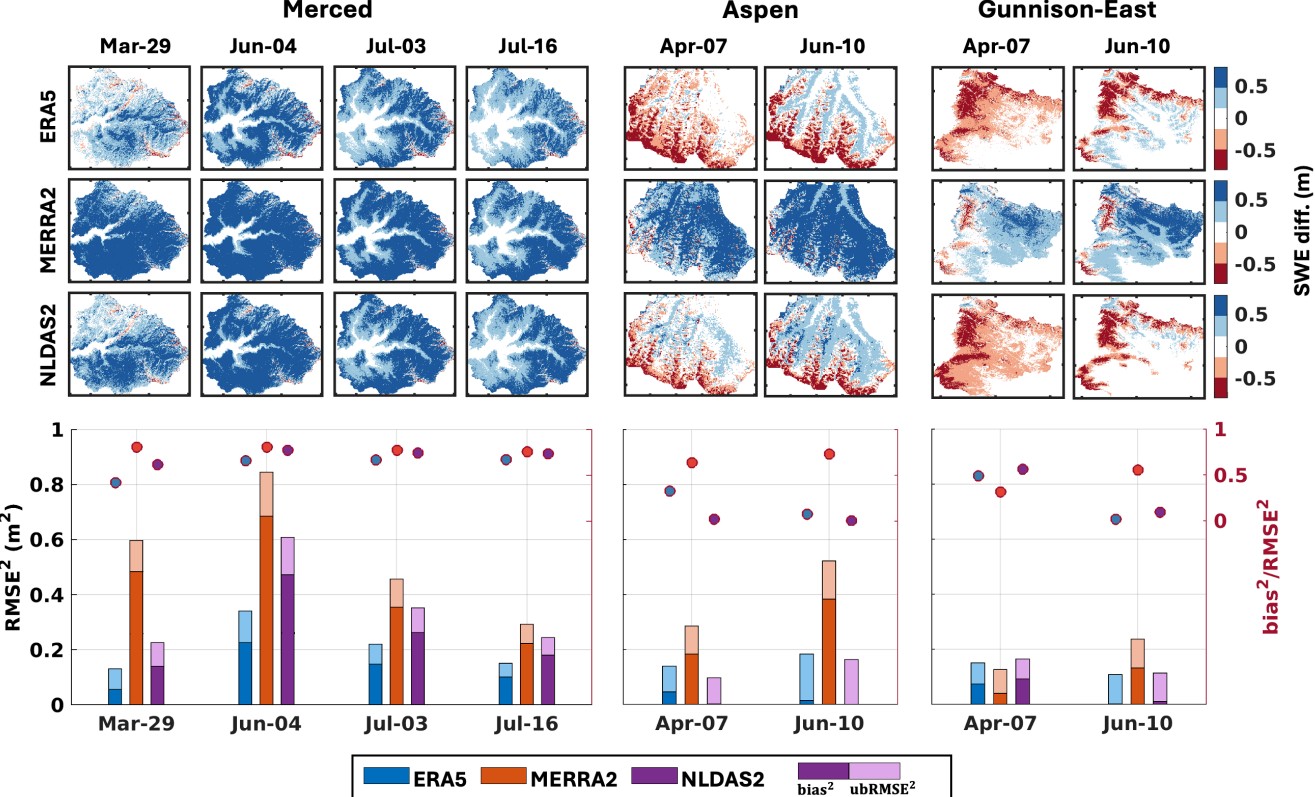

**Figure 6.** Spatial maps of the difference between prior mean SWE (m) and ASO-based SWE (m) at each ASO observation time across the study domains (top panel). Stacked bar plot of $RMSE^2$ for the prior mean SWE, compared to ASO-based SWE (bottom panel). The ratio of $bias^2$ to $RMSE^2$ is represented by colored dots, referenced to the right y axis.

## 3.2 Experiment 1b: Value of multi-forcing ensemble on model-based (prior) SWE

The evaluation of single-forcing prior mean SWE was used to inform the multi-forcing ensemble design. Specifically, the number of realizations drawn from each forcing dataset is determined by partitioning weights based on RMSE relative to ASO-based SWE near April 1st. Peak SWE was treated as the key target variable, assuming that errors at this time are indicative of the errors over the seasonal cycle across the datasets. The partitioning weight of each forcing dataset and corresponding number of realizations are calculated using Eqs. 1-6 and summarized in Table 3. The value of the multi-forcing ensemble for model-based prior SWE is evaluated by comparing its accuracy to that of individual forcing datasets (Experiment 1a). Accuracy is assessed against ASO-based SWE using RMSE, absolute bias, and ubRMSE.




**Table 3.** Partitioning weights (*W*) and number of realizations used in the multi-forcing ensemble for each domain based on the RMSE of prior mean SWE relative to ASO-based SWE near April 1ˢᵗ.

| Domains | | ERA5 | MERRA2 | NLDAS2 |
|---|---|---|---|---|
| Merced | *W* | 0.55 | 0.13 | 0.33 |
| | # of realizations | 66 | 15 | 39 |
| Aspen | *W* | 0.34 | 0.17 | 0.49 |
| | # of realizations | 41 | 20 | 59 |
| Gunnison-East | *W* | 0.33 | 0.38 | 0.29 |
| | # of realizations | 39 | 46 | 35 |


Figure 7 shows these metrics for the worst-performing and best-performing single-forcing cases, as well as the multi-forcing case. Overall statistics are calculated with aggregated results across all times and domains. The multi-forcing case consistently outperforms the worst-performing single-forcing case. The multi-forcing case shows significant reductions in RMSE and absolute bias, outperforming even the best-performing single-forcing dataset. This improvement results from the balancing of

systematic errors across individual forcing datasets. For example, MERRA2 tends to exhibit positive biases, while ERA5 and NLDAS2 have negative biases. When combined in the multi-forcing ensemble, such opposing errors span the truth and partially offset, yielding improved accuracy in the ensemble mean. The overall statistics further illustrate the potential for improved performance by a multi-forcing approach for large spatial scale modeling applications, where different forcing datasets are likely to have different spatial error characteristics (including bias). Specifically, the multi-forcing case achieves

a lower RMSE (0.47 m) than MERRA2 (0.63 m) and NLDAS2 (0.51 m), though slightly higher than ERA5 (0.43 m). For absolute bias, the multi-forcing case significantly reduces errors to 0.24 m, compared to 0.49 m for MERRA2. In terms of ubRMSE, the multi-forcing case performs comparably to the others, with an overall value of 0.40 m – lower than NLDAS2 (0.48 m) and ERA5 (0.42 m) and close to MERRA2 (0.39 m). The multi-forcing approach effectively reduces prior mean SWE errors, including both systematic (absolute bias) and random (ubRMSE) errors, across various snow conditions. The

improvements are more significant in terms of the absolute bias and RMSE, whereas the ubRMSE is more comparable across different cases.



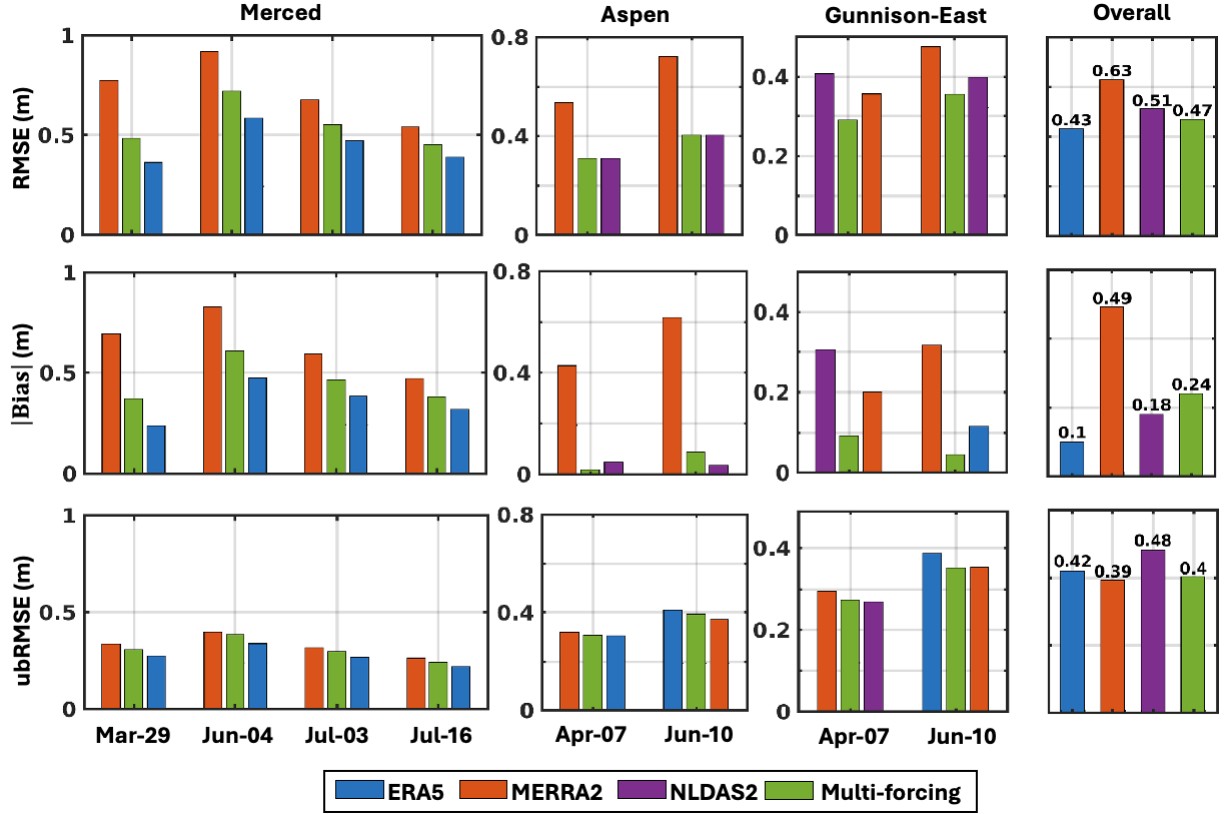

**Figure 7.** Bar plots of the RMSE, absolute bias, and ubRMSE of prior mean SWE compared to ASO-based SWE across all domains and ASO observation times. For each date and domain, the three bars show the worst-performing single-forcing, the multi-forcing, and the best-performing single-forcing, respectively. The rightmost column summarizes the overall statistics across all domains and dates.

## 3.3 Experiment 2a: Impact of single dataset meteorological forcings and assimilation of near-peak snow depth on posterior SWE estimation

The following analysis evaluates the impact of assimilating SD (near ~April 1st) on SWE accuracy by comparing prior and posterior ensemble mean estimates. Assimilation at this time within the water year corrects near-peak SWE and the errors accumulated during the snow accumulation season. Prior SWE accuracy was assessed in Sect. 3.1, and here we focus on how assimilation reduces both systematic and random components of those errors. Importantly, the reduction in errors on the assimilation day can be interpreted as largely attributable to correction of accumulation-season forcing errors, since the assimilation effectively "resets" SWE to near-true values. Assimilating near-peak ASO SD therefore constrains posterior SWE errors to a likely lower bound, with subsequent deviations reflecting primarily melt-season forcing uncertainties, as accumulation-season biases have been corrected. Figure 8 illustrates the reduction in SWE errors from prior to posterior mean





estimates for each study domain. The posterior estimates significantly and consistently reduce both systematic (bias) and random (ubRMSE) errors compared to the prior. On the assimilation day, posterior RMSE values drop from ~0.4 – 0.6 m in the prior to <0.1 m across all domains, with absolute bias reduced to near 0 m. During the snowmelt season (e.g., in June-July),

posterior SWE errors are still lower than those of the prior estimates, with RMSE typically reduced by more than 50%. This improvement primarily reflects the reduction of accumulation-season error propagation and isolates the remaining errors to those specific to melt-season processes. However, posterior SWE accuracy during the snowmelt period declines with time after the assimilation date, as expected, due to snowmelt error propagation from the forcing datasets. Moreover, ubRMSE becomes the dominant component of posterior RMSE, larger than the absolute bias across all domains during both the

assimilation time and snowmelt period. Additionally, the narrower spread in posterior error distributions (i.e., RMSE and absolute bias), compared to the broader range in prior estimates, indicates that the SD data assimilation effectively reduces errors to a similar extent across different meteorological forcings.

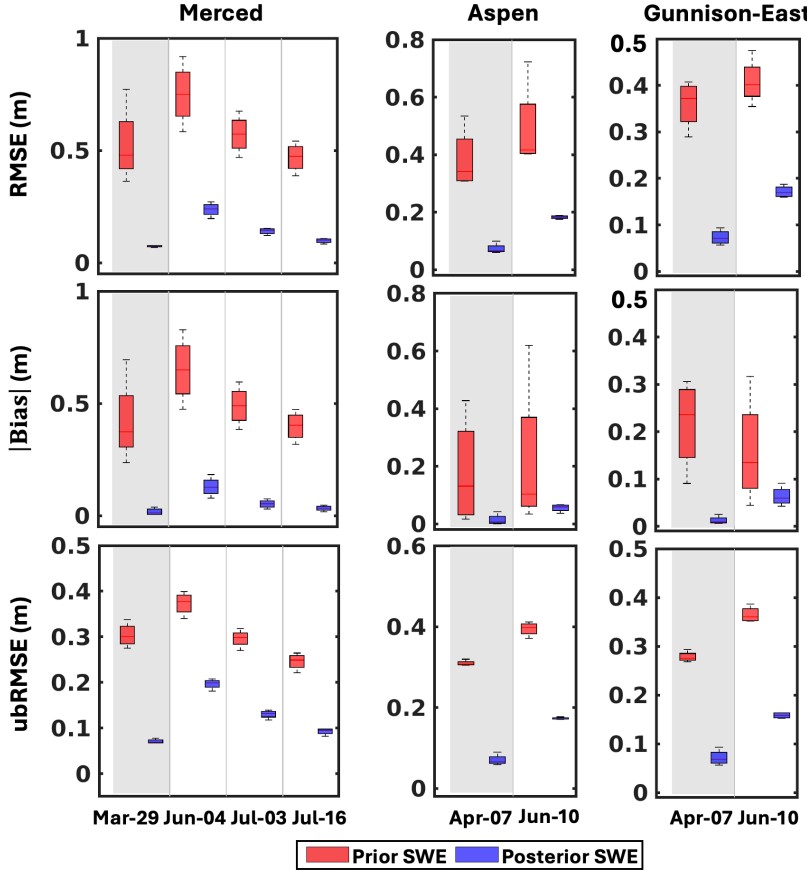

**Figure 8.** Box plots of the RMSE, absolute bias, and ubRMSE for prior (red) and posterior (blue) mean SWE estimates

compared to ASO-based SWE across all domains and ASO observation times. Each box represents the distribution of error





metrics across all single-forcing cases (ERA5, MERRA2, and NLDAS2) and the multi-forcing case for a date and domain. The light grey shaded area highlights the assimilation date.

The prior and posterior errors shown in Fig. 8, can be used to gain some insight into the relative impact of forcing error: (1) during the accumulation-season on peak SWE vs. (2) after peak SWE on melt-season SWE (Table 4). Specifically,
the reduction in RMSE from prior to posterior on the assimilation day (ranging from 0.29 to 0.45 m) provides a metric that is largely representative of the total error accumulated during the winter, for which uncertainty in snowfall is a primary driver. Once that accumulation-season error is removed, by assimilating SD near peak SWE, the residual posterior RMSE after peak SWE (ranging from 0.16 to 0.18 m) is more indicative of errors introduced during the melt period, such as those from melt flux forcings or internal model physics (e.g., albedo representation). This comparison indicates that winter accumulation
forcings likely contribute on the order of twice as much SWE error as melt-season forcings, making them the dominant source of uncertainty in the model.

**Table 4**. Average SWE RMSE across all forcing scenarios on the assimilation date (near peak) and during the melt season. The reduction in near-peak RMSE (prior-posterior) quantifies corrected accumulation season errors (attributable mostly to
forcing-driven snowfall errors), while the residual posterior RMSE represents melt season errors (attributable mostly to forcing-driven melt flux errors).

| Domain | Prior near peak SWE RMSE on assimilation day (m) | Posterior near-peak SWE RMSE on assimilation day (m) | Near-peak SWE (accumulation season) RMSE reduction (m) | Posterior melt-season (residual) SWE RMSE (m) |
|---|---|---|---|---|
| Merced | 0.52 | 0.07 | 0.45 | 0.16 |
| Aspen | 0.38 | 0.07 | 0.31 | 0.18 |
| Gunnison-East | 0.36 | 0.07 | 0.29 | 0.17 |

## 3.4 Experiment 2b: Value of multi-forcing ensemble on DA-based (posterior) SWE accuracy

The comparison of posterior mean SWE accuracy for the multi-forcing case against the worst- and best-performing
single forcing cases across all dates and study domains is shown in Fig. 9. Consistent with the pattern observed in the prior SWE comparison (Fig. 7), the multi-forcing case consistently outperforms the worst-performing single-forcing scenario across all domains and dates. Note that during the snowmelt season, MERRA2-based posterior SWE estimates exhibit the highest accuracy across all domains, indicating that the assimilation of ASO SD near April 1st effectively corrects prior SWE errors – largely due to high systematic bias in MERRA2, and the fact that its temporal SWE evolution better aligns with ASO-based
SWE compared to ERA5 and NLDAS2. This indicates that the worst-performing prior does not necessarily result in the worst-performing posterior, and vice versa. While prior performance is critical for modeling without data assimilation, it does not directly determine posterior performance when data assimilation is applied.

Across all metrics, the differences in posterior SWE accuracy among forcing datasets are relatively small. The multi-forcing case yields a slightly lower RMSE (0.14 m) than both NLDAS2 (0.16 m) and ERA5 (0.15 m). Absolute bias is low



across all cases, with the multi-forcing case (0.04 m) showing marginal improvement over NLDAS2 and ERA5 (both 0.05 m). In terms of ubRMSE, the multi-forcing case (0.13 m) has a comparable performance to the best single-forcing case (MERRA2) and outperforms both ERA5 (0.14 m) and NLDAS2 (0.15 m). Across all cases, posterior SWE errors are dominated by the ubRMSE, which is higher than absolute bias. While the overall statistics show that the multi-forcing approach leads to slightly improved posterior SWE accuracy, the differences among cases are relatively small. Without data assimilation, the multi-forcing ensemble provides greater value by mitigating the impact of large prior errors from any single forcing dataset, particularly when the forcing errors are unknown. However, in the data assimilation context examined here, much of the prior error is corrected through the assimilation of ASO snow depth, which provides a direct constraint on SWE, thereby reducing the overall error range across single-forcing cases and thus reducing the added benefit of the multi-forcing approach in improving spatial accuracy.

The posterior accuracy of the multi-forcing case does not exceed that of the best single-forcing case. This is partly because the ensemble size is fixed at 120 realizations, and distributing this across multiple forcing datasets can dilute the contribution from the best-performing dataset and degrade the performance. Additionally, the ensemble weights were selected to minimize *prior* peak SWE errors, which will not necessarily align with minimizing posterior SWE errors after assimilation. While MERRA2 yields the most accurate posterior SWE estimates at the study sites for WY 2019, the difference in performance compared to other forcings is relatively small. The potential penalty for selecting a suboptimal forcing is relatively minor when SD data assimilation is applied. This is in contrast with modeling-only applications, where errors across forcings are more variable and the consequences of choosing a less accurate forcing can be more significant. Further evaluation is needed to determine whether MERRA2 consistently performs best across broader spatial and temporal domains.





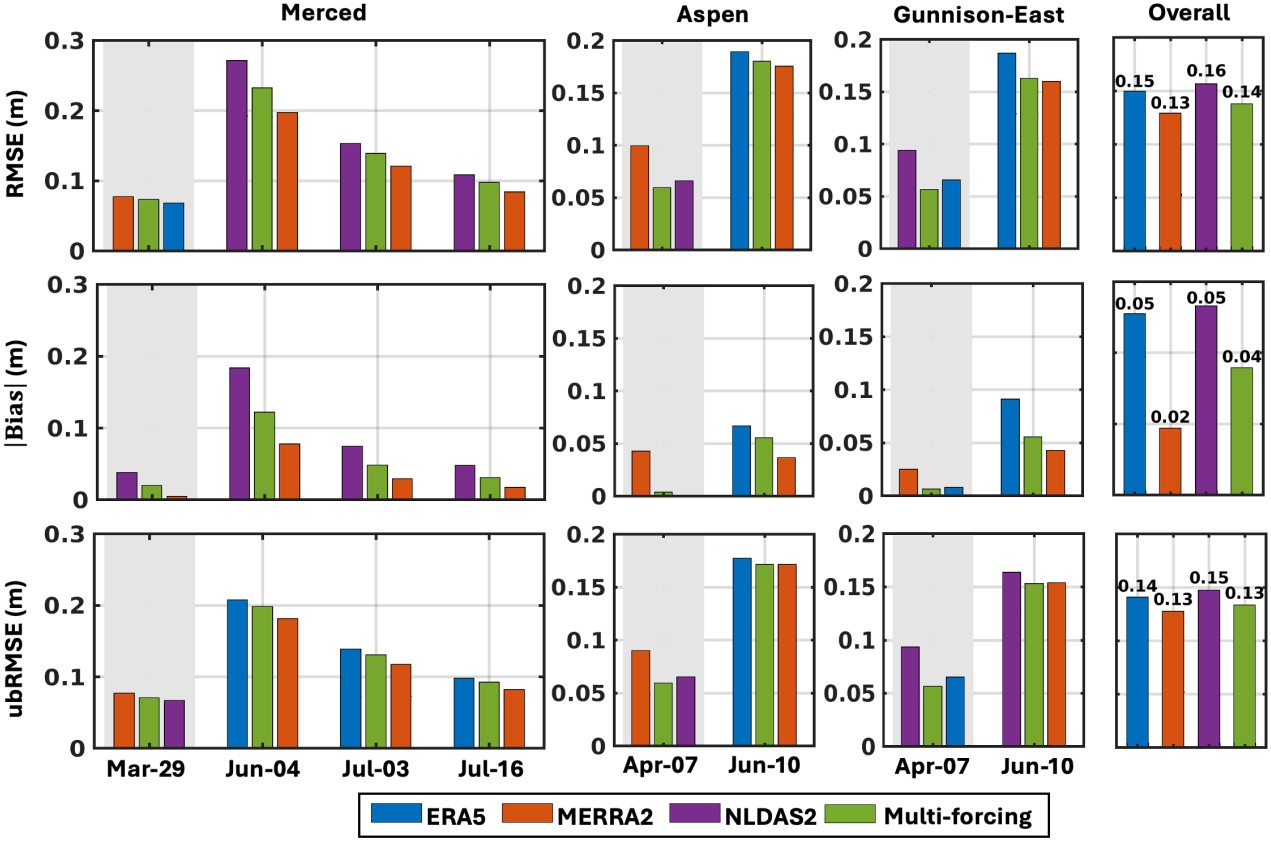

**Figure 9.** Same as Fig.7, but for the posterior mean SWE. The added light grey shaded area highlights the assimilation date.

### 3.5 Experiment 2b: Value of multi-forcing ensemble on DA-based (posterior) SWE uncertainty characterization

While the above analysis focuses on the posterior mean, characterizing uncertainty is also important. The magnitude of posterior uncertainty is influenced by the assimilation method, the observational data, and its measurement error characteristics. In this study, we focus on relative differences in posterior uncertainty across cases. A more diverse set of inputs may help reduce the risk of overconfident estimates, particularly in methods subject to degeneracy, such as particle-based approaches. In this study, posterior SWE ensemble spread is defined as the posterior mean ($\mu$) $\pm$ 2 times the standard deviation ($\sigma$). To evaluate this uncertainty characterization, we examine whether the defined posterior ensemble spread captures the ASO-based SWE. Assuming that the assimilation of ASO SD near April 1st provides accurate constraints on both the mean and uncertainty of SWE at the assimilation time, our analysis focuses on the snowmelt period, when June ASO observations (which were not assimilated) are available and differences among forcing scenarios are more likely to emerge.

Figure 10 displays the fraction of watershed area where the posterior SWE spread ($\mu \pm 2\sigma$) captures the ASO-based SWE during the snowmelt period in June. The spatial coverage metric indicates the proportion of grid cells within each watershed where the ASO-based SWE lies within the posterior SWE uncertainty range. While an ideal assimilation system




might achieve near-complete coverage (approaching 100%), the maximum coverage observed is around 70%. However, the

focus of this analysis is not on the absolute coverage values, but rather on the relative differences between single-forcing and multi-forcing experiments. These qualitative patterns are expected to hold even if a different data assimilation method were used. Among all single-forcing scenarios, MERRA2 exhibits the highest coverage, with values of 0.56, 0.63, and 0.69 over Merced, Aspen, and Gunnison-East, respectively. In contrast, ERA5 yields lower coverage fractions of 0.46, 0.47 and 0.48, while NLDAS2 yields values of 0.31, 0.50, and 0.53. The multi-forcing case yields improvements in the uncertainty

characterization with higher coverage fraction than the poor-performing single-forcing case. For example, in Merced, using NLDAS2 alone results in only 31% of pixels being within the predicted uncertainty, meaning that 69% of the watershed area has ASO-based SWE outside the posterior uncertainty range. The multi-forcing case, however, achieves 49% coverage, reducing the failure rate to 51%. Similarly, in Aspen and Gunnison-East, the multi-forcing yields coverage of 52% and 63%, respectively. These values improve upon the lower-performing single-forcing cases but do not reach the best-performing case.

Note that while MERRA2 performs best in this specific analysis, this may not generalize across regions. When the best individual forcing at a given location is unknown a priori, the multi-forcing provides a relatively robust alternative. Although it does not outperform the best single-forcing case, it consistently performs better than the worst and, in this case, outperforms two out of three datasets. The multi-forcing coverage also tends to reflect the influence of the dominant partitioning weights in each domain, especially under fixed ensemble size constraints. For instance, the coverage is closest to ERA5 in Merced,

NLDAS2 in Aspen, and MERRA2 in Gunnison-East, which correspond to the forcing dataset that has the largest partitioning weight in each domain. This pattern suggests that the posterior SWE spread in each watershed reflects the uncertainty characteristics of the most influential forcing dataset. This highlights the importance of multi-forcing ensemble design and partitioning weight optimization for improving uncertainty characterization in snow reanalysis.

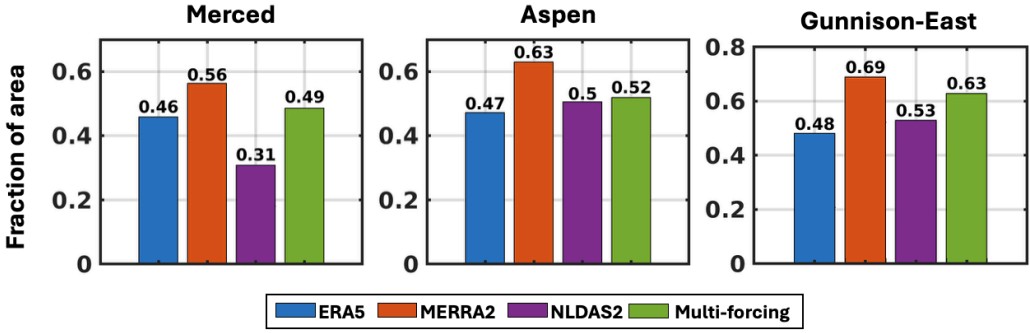

**Figure 10.** Fraction of watershed area where the posterior SWE ensemble spread captures ASO-based SWE during the snowmelt period, when June ASO observations are available.

Figure 11 shows maps illustrating the performance of the posterior SWE uncertainty characterization, defined as the number of single-forcing posterior SWE uncertainty characterizations that do not capture the ASO-based SWE during the

snowmelt period in June, categorized from 0 to 3. Each pixel is classified based on how many of the three single-forcing cases



(ERA5, MERRA2, and NLDAS2) miss the ASO-based SWE within their respective uncertainty ranges. Across all three domains, a large portion of the watershed area falls into two categories: either all three ensembles capture the ASO-based SWE or none of them do. In regions with three misses, covering 26% to 36% of the area, the multi-forcing ensemble has a near-zero probability of capturing the ASO-based SWE, indicating, as expected, limited improvement from combining poor-performing inputs. Conversely, in areas where all three single-forcing ensembles succeed in uncertainty characterization (24 – 40% of the watershed area), the multi-forcing ensemble captures the ASO-based SWE with near 100% success.


The greatest value of the multi-forcing approach is illustrated in cases where one or two of the single-forcing uncertainty ranges miss the ASO-based SWE. For example, when only one single-forcing ensemble misses, the multi-forcing ensemble spread captures the ASO-based SWE in over 60% of pixels. Even in more challenging cases where only one single-forcing uncertainty characterization succeeds, the multi-forcing spread still captures the ASO-based SWE in roughly 25% to 50% of pixels. The multi-forcing approach can effectively improve uncertainty characterization in regions where at least one single-forcing provides accurate estimate of SWE uncertainty.


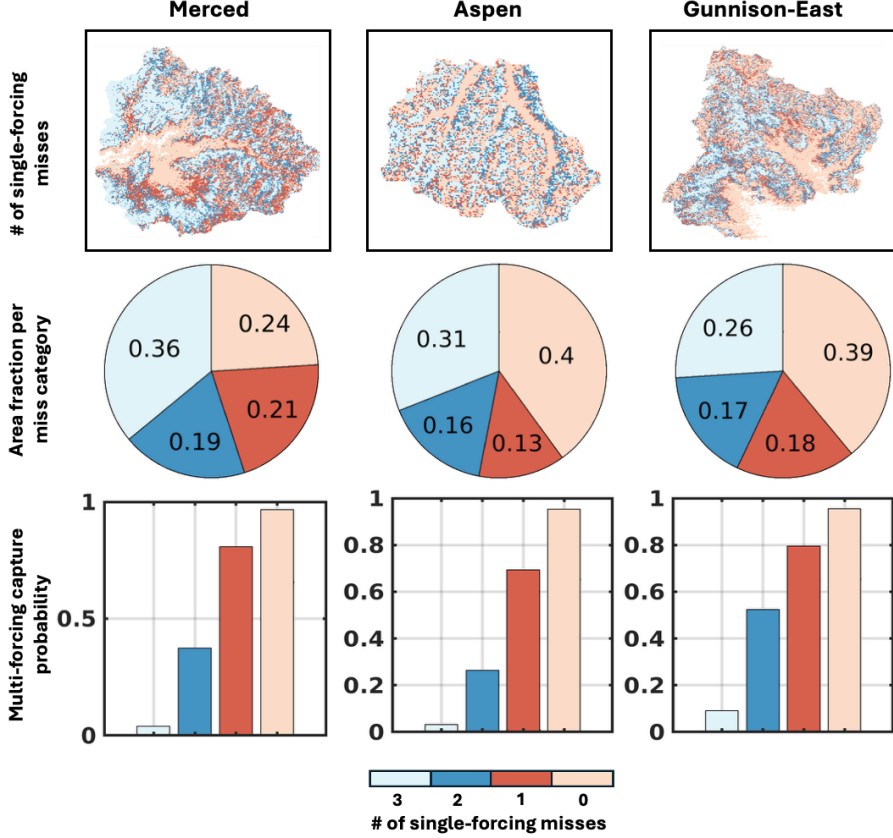

**Figure 11.** Classification maps illustrating the number of cases where the single-forcing posterior SWE ensemble spread misses the ASO-based SWE during the mid-melt time in June (top panel). Pie charts of the fraction of watershed area corresponding to each category of posterior miss count (middle panel). Bar plots (bottom panel) show the conditional






probability that the multi-forcing posterior spread captures ASO-based SWE, calculated as the number of pixels where the multi-forcing spread captures ASO-based SWE divided by the total number of pixels within each category.

**3.6 Sensitivity analysis of the multi-forcing partitioning weights**

The partitioning weights used in the previous analysis were derived using lidar-based SWE, which provide high-resolution spatial coverage but are not typically readily available in most regions. In areas lacking such spatially distributed reference data, determining optimal weights for combining multiple forcings becomes more challenging. This limitation motivates the need to explore whether simpler or more broadly applicable weighting methods can still yield improvements. In practice, neither prior nor posterior SWE errors are known in advance, and the two may differ substantially as shown above.

This uncertainty has implications for both prior modeling and data assimilation, as different weighting strategies may be more appropriate depending on the context. To evaluate the influence of weighting strategies, we conducted a sensitivity analysis examining their impact on prior and posterior SWE accuracy and the characterization of posterior uncertainty. The simplest approach is to assign equal weight to all forcing datasets, assuming no prior knowledge of their relative performance. While this approach is applicable in data-sparse regions, it may degrade accuracy compared to optimally weighted combinations by

equally incorporating errors from less accurate datasets. Alternatively, weights can be determined based on the relative performance of each dataset, using random error metrics such as the ubRMSE. Recent studies have demonstrated the feasibility of triple collocation analysis (Yilmaz et al., 2012; Yoon et al., 2019) for estimating unbiased error variances without requiring ground truth, offering a potential path for weight estimation in ungauged regions. However, applying triple collocation to snow reanalysis has some challenges. For example, (1) Systematic differences among datasets can bias SWE estimates, where biases

are not accounted for by standard triple collocation; and (2) triple collocation methods have been applied to individual variables separately, where different variables within the same dataset (e.g. ERA5 precipitation and air temperature) could have divergent weights.

In this analysis, we tested three simple partitioning strategies with different assumptions about available knowledge of the single-forcing performance: (1) uniform weighting across datasets (Uniform), (2) weighting based on prior SWE errors

measured by ubRMSE (ubRMSE-based), which represents the relative performance of each dataset based on random error, and (3) weighting based on prior SWE errors measured by RMSE (RMSE-based), consistent with approaches used in Experiments 1b and 2b. The first is simplest and assumes all forcing datasets are of equal quality, with no prior information on their performance; the second assumes that ubRMSE could be derived from techniques like triple collocation without a reference dataset; the third is theoretically optimal but requires knowledge of bias and ubRMSE (and may not be optimal for

the posterior estimates as shown above). Both ubRMSE and RMSE were computed by comparing prior mean SWE to ASO-based SWE near April 1st, and the resulting weights are summarized in Table 5. To assess the influence of each weighting strategy, we quantified changes in SWE accuracy and uncertainty representation relative to the worst-performing single-forcing case. Specifically, we computed the maximum relative improvement in prior and posterior mean SWE accuracy, defined as:



$$I_{accuracy} = -\frac{M - W}{W}$$
Eq. 7

where M and W are the domain- and time-aggregated error metrics (i.e., RMSE, absolute bias, and ubRMSE) for the multi-forcing and worst single-forcing cases, respectively. Similarly, the relative improvement in posterior SWE uncertainty is calculated as:

$$I_{uncertainty} = \frac{F_M - F_W}{F_W}$$
Eq. 8

where $F_M$ and $F_W$ represent the fraction of the area where the posterior ensemble uncertainty captures the ASO-based SWE in in June (snowmelt season), for the multi-forcing and worst single-forcing cases, respectively.

**Table 5.** Partitioning weights (*W*) used in the multi-forcing ensembles.

| Domain | Strategy | ERA5 | MERRA2 | NLDAS2 |
|---|---|---|---|---|
| Merced | Uniform | 0.33 | 0.33 | 0.33 |
|  | ubRMSE-based | 0.39 | 0.26 | 0.35 |
|  | RMSE-based | 0.55 | 0.13 | 0.32 |
| Aspen | Uniform | 0.33 | 0.33 | 0.33 |
|  | ubRMSE-based | 0.34 | 0.32 | 0.34 |
|  | RMSE-based | 0.34 | 0.17 | 0.49 |
| Gunnison-East | Uniform | 0.33 | 0.33 | 0.33 |
|  | ubRMSE-based | 0.34 | 0.30 | 0.36 |
|  | RMSE-based | 0.33 | 0.38 | 0.29 |

585        As shown in Fig. 12, all partitioning weight scenarios improve the prior mean SWE accuracy relative to the worst single-forcing case, with relative improvements ranging from 44% to 51% in absolute bias, 18% to 25% in RMSE, and 9% to 16% in ubRMSE. The RMSE-based case yields the largest improvement, which is expected given its design to minimize the prior SWE error. The ubRMSE-based case performs slightly better than the Uniform case in reducing prior SWE errors. For posterior mean SWE accuracy, all three weighting cases show comparable performance, with the maximum relative 590 improvements ranging from 33% to 38% in absolute bias, 12% to 14% in RMSE, and 10% to 12% in ubRMSE. These improvements are smaller than those observed for the prior mean SWE, suggesting that the assimilation of near-peak ASO SD reduces the sensitivity of posterior SWE accuracy to the partitioning weights, consistent with findings in Sect. 3.4. Among the strategies, the Uniform case achieves the highest improvement in posterior SWE accuracy, though the differences among all three cases are marginal. This is likely due to both performance-based schemes assigning relatively low weights to MERRA2, 595 which has high prior errors but the most accurate posterior estimates after assimilation.

        All three multi-forcing scenarios also improve the posterior SWE uncertainty characterization, as indicated by increased fractions of area where ensemble spread captures ASO-based SWE in June: 58% to 74% in Merced, 10% to 21% in Aspen, and 29% to 31% in Gunnison-East. The Uniform case provides the largest improvement in uncertainty, likely because it assigns more weight to MERRA2, which has the most accurate posterior uncertainty characterization (Fig. 10). Overall, no 600 single weighting strategy consistently outperforms the others across all evaluation metrics. The RMSE-based strategy provides



the largest improvement in prior SWE accuracy, as it is optimized to reduce prior errors. However, differences in posterior SWE accuracy among the weighting strategies are relatively small. This indicates that when DA is able to correct most of the prior errors, the sensitivity of posterior accuracy to the partitioning weights is significantly reduced. The Uniform case shows marginally better performance in posterior SWE accuracy and uncertainty characterization across the three study domains in WY 2019. This is likely because it assigns greater weight to MERRA2, which, despite having higher prior errors, contributes positively to posterior estimates after assimilation. Using multiple forcing datasets, regardless of partitioning strategy, improves both prior and posterior SWE estimates relative to the worst-performing single-forcing dataset across all study domains. The value of different weighting strategies may depend on the DA performance and the characteristics of the forcing data.

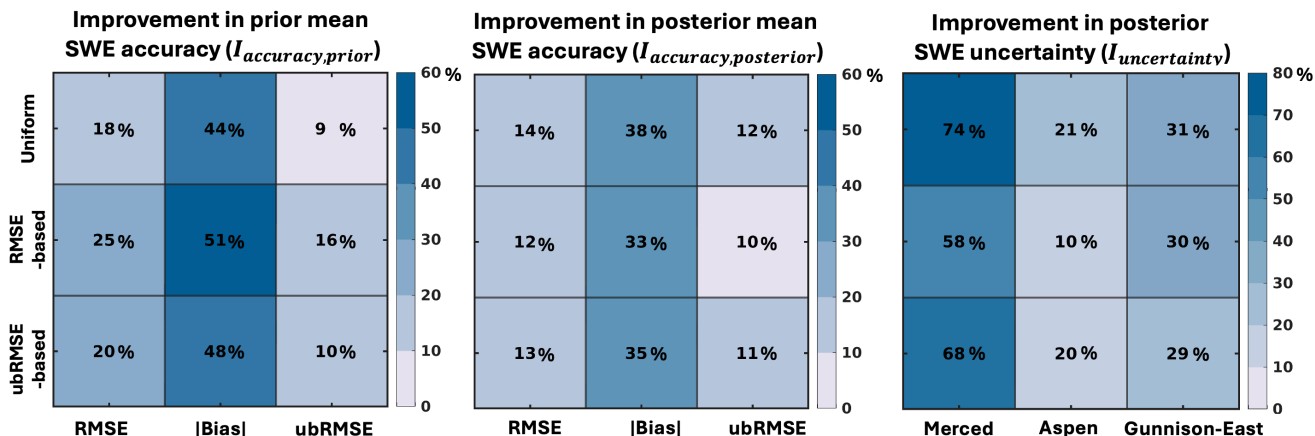

**Figure 12.** Maximum relative improvements from multi-forcing scenarios with different partitioning strategies. Left and middle panels show improvements in RMSE, absolute bias, and ubRMSE for prior and posterior mean SWE. The right panel shows improvement in the fraction of area where posterior ensemble spreads capture ASO-based SWE in June.

## 4 Conclusions

This study examined the impact of different meteorological forcings on both model-based (prior) and DA-based (posterior) ensemble SWE estimates and explored whether integrating multiple forcing datasets provides added value. We implemented an ensemble Bayesian framework that includes both prior and posterior estimates over case study domains where high-resolution, lidar-based snow observations are available for both assimilation and analysis.

The prior modeling analysis showed significant variability in peak SWE estimates, primarily driven by differences in cumulative snowfall. These differences propagate into the ablation season and vary depending on the meteorological forcing dataset used. No single-forcing prior SWE consistently outperforms the others across all study domains, and the accuracy ranking of prior SWE estimates varies with elevation. The errors in prior SWE were further decomposed into bias and unbiased root mean square error (ubRMSE). Differences in prior SWE errors among datasets are largely dominated by variations in




bias, which are greater than those in ubRMSE. In poorly performing single-forcing datasets, bias contributes more significantly
to the total error.

The evaluation of single-forcing prior SWE informs the design of the multi-forcing ensemble tested herein, in which partitioning weights for each forcing dataset are determined using a least square weighting approach. Forcing datasets that yield more accurate prior SWE are assigned higher weights than those with lower performance. Across all regions, the multi-forcing ensemble consistently outperforms two out of the three single-forcing cases, significantly in terms of RMSE and bias,
while ubRMSE remains more comparable among the cases. This suggests that differences in prior SWE performance are primarily driven by biases in the forcing inputs, whereas the random error components are more consistent across datasets.

The assimilation of a single lidar-based snow depth observations near April 1$^{st}$ corrects a substantial portion of prior SWE errors and helps isolate accumulation-season forcing errors from those specific to the melt process. DA reduces the impact of forcing uncertainties, resulting in smaller differences in posterior SWE accuracy compared to the prior, primarily up
to peak SWE. As spatially variable biases are corrected through assimilation, ubRMSE becomes the dominant component of posterior RMSE throughout the year. Assimilation of snow depth narrows the error range of overall SWE across single-forcing cases, reducing the added value of the multi-forcing approach for improving spatial accuracy. This may be less true if another variable that is less directly connected to peak SWE were assimilated (e.g., fractional snow-covered area (fSCA)). Therefore, the influence of forcing errors and the potential added value of a multi-forcing ensemble could vary depending on the type of
observation assimilated. However, the multi-forcing ensemble in the SD assimilation case study still provides marginal improvements in posterior SWE accuracy and outperforms two of the three single-forcing datasets. Its uncertainty reflects the characteristics of the most influential forcing dataset, defined by the largest partitioning weight. This approach improves the robustness of uncertainty characterization, especially when at least one forcing dataset provides a reliable uncertainty estimate.

Sensitivity tests of the partitioning weights used in the multi-forcing ensemble suggest that using multiple forcing
datasets, regardless of the specific weighting strategy, can improve both prior and posterior SWE estimates when compared to the worst-performing single forcing dataset across all study domains. The accuracy of prior SWE is more sensitive to the choice of partitioning weights, with those based on RMSE of peak SWE providing the greatest added value, as expected. Although this case study relies on lidar data that is not widely available, the insights are broadly applicable. In regions with limited data, modeling and data assimilation frameworks that incorporate multiple forcing datasets (even using simple equal
weighting) can still produce more reliable SWE estimates. It is generally difficult to know a priori which input dataset is "best" or "worst", and in practice, no single dataset is likely to be universally best across space and time. A multi-forcing approach is therefore likely to be beneficial as it hedges against use of any single input. Developing methods that leverage the strengths of multiple datasets is both scientifically justified and operationally wise. This study provides one clear example of the benefits of such an approach.



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
