# Peer review of "Assessing the impact of meteorological forcing and its uncertainty on snow modeling and reanalysis"

_EGUsphere, 2025_

## Referee Comment (RC1)

**Revisions of: Assessing the impact of meteorological forcing and its uncertainty on snow modeling and reanalysis**
**By: Sun and Margulis**

Benoit Montpetit

**1 General Comments**

This paper aims at analyzing the impact of forcing data variability on ensemble SWE estimates. Here, the objective is clearly not to demonstrate which forcing dataset is best overall, but rather show that by using multiple products, the multi-forcing ensemble show better performances than single-forcing ensembles separately. I really appreciated reading this manuscript, since this question of "which forcing dataset to use" is still up for debate in the cryospheric community and clearly, there is no "one-size fits all" answer yet. This paper is a step in the right direction to potentially standardize forcing datasets used for large domain studies.

The manuscript is well written and structured. The analysis structure with the four different experiments also helps readers to understand the impacts of the different forcing datasets and the impact of assimilating observations on the different forcings. Even though there are different Bayesian frameworks and downscalling methods that could perform better, I feel like this is not the objective of this study and the selection of the methodology is well described here, and well documented in previous publications, showing that it is applicable to this context.

That said, I do have some questions with regards to experiment 1a, where I feel like some information in missing to properly characterize where the variability of the different forcings and their uncertainties come from. This will be detailed in the next section.

Even though I call the latter "major comment", I still recommend the publication of this manuscript after what I consider minor revisions.

**2 Major Comments**

With the three watersheds included in this study and the differences in spatial resolution of the original input datasets, I feel like there should be a bit more information included, which could benefit and strengthen the discussion in section 3.1.

The first information to consider is land cover type for the different watersheds. Not being extremely familiar with all watersheds, it would be useful to know the different and dominant land cover types for each. Knowing that there are uncertainties linked to land cover type, this could help indicate why one forcing datasets might perform better for one watershed and not the other.

In combination with land cover, knowing the initial altitude to which the surface properties are given for the different datasets could help identify sources for the SD/SWE variability. Again, there are uncertainties in modelled SWE associated to its altitude.

Both information also provide discussion points with regards to rescaling the dataset to 150 m. I feel like this could explain some of the outputs we see in annual snowfall of Figure 2.

As mentioned by the authors, the objective of this addition is not to identify which forcing dataset is best, but in a context of analyzing various domains, this could indicate the importance of which datasets to use in a multi-forcing experiment.

**3 Minor Comments**

Figure 1: Would be interesting to add the land cover map, which is used in resampling (Figure 3).

Eqns. 1-3: Personally, I prefer the more compact ways of writing these equations. They could be shorten into one equation:

$$W_i = \frac{\prod_{j \neq i} \sigma_j^2}{\sum_i \prod_{j \neq i} \sigma_j^2} \tag{1}$$

Eqns. 4-6: Same as the previous equations, it could be summarized by:

$$N_i = W_i \times N \tag{2}$$

where $N = \sum_i N_i$

L. 321: Please clarify what is meant by "SWE variations". I assume it is the difference with ASO SWE observations.

40 Figure 4: it would be interesting to add error bars on the ASO measurements to show the spatial variability of SD and SWE across the different study areas and also the spread in SWE due to the mean ensemble density estimation. You show which forcing data works better on average, but the error bars could indicate whether the different forcing datasets provide estimates within the range of measured properties.

Table 3: Probably a precision error but weights do not add up to 1 for Merced.

45 Figure 7: I understand the idea behind keeping only the best and worst cases in this figure but I think it would still be relevant to include all three forcing cases. I would also keep the same order of presentation in the figure as previous figures to make it easier to compare. I also like to keep the y-axis range the same when comparing plots horizontally. It helps to identify which site has larger errors more easily.

---

## Author Comment (AC1)

We would like to thank the reviewers for their thorough reviews and constructive comments on the manuscript. The original comments are shown in regular black font. The responses to reviewer comments are shown in blue font, with text describing proposed additions and revisions of the manuscript shown in red font. Any original manuscript text is shown in gray font. Figures included in this response document that are labeled as "Figure. Rx" are provided for clarification only. Figures proposed for inclusion in the revised manuscript are labeled as "Figure x" or as supplementary figures "Figure. Sx".

**Reply to Reviewer #1 Comments**

**MAJOR COMMENTS**

With the three watersheds included in this study and the differences in spatial resolution of the original input datasets, I feel like there should be a bit more information included, which could benefit and strengthen the discussion in section 3.1.

The first information to consider is land cover type for the different watersheds. Not being extremely familiar with all watersheds, it would be useful to know the different and dominant land cover types for each. Knowing that there are uncertainties linked to land cover type, this could help indicate why one forcing dataset might perform better for one watershed and not the other.

In combination with land cover, knowing the initial altitude to which the surface properties are given for the different datasets could help identify sources for the SD/SWE variability. Again, there are uncertainties in modelled SWE associated to its altitude.

Both information also provide discussion points with regards to rescaling the dataset to 150 m. I feel like this could explain some of the outputs we see in annual snowfall of Figure 2.

As mentioned by the authors, the objective of this addition is not to identify which forcing dataset is best, but in a context of analyzing various domains, this could indicate the importance of which datasets to use in a multi-forcing experiment.

Response:

We appreciate the reviewer's suggestion to provide additional information on land cover characteristics and the elevation of the forcing datasets, and to clarify how these factors influence the spatial variability of snowfall and SWE across the three watersheds.

(1) Land cover type and forest fraction: we propose to add a new figure showing land cover type and fractional forest cover for the three study watersheds (Fig. R1). Land cover and forest fraction are prescribed as static model inputs for all experiments. Each forcing uses the same datasets – AVHRR for land cover (1 km) and GLCF for forest cover (30 m) – interpolated to the 150 m model grid using nearest neighbor interpolation. They are identical across forcing datasets within a given watershed but differ between watersheds.

Figure S1 indicates significant differences in dominant land cover among the three domains. Merced is characterized primarily by grassland and wooded grassland, Aspen exhibits a mix of deciduous forest at lower elevations transitioning to wooded grassland and alpine terrain at higher elevations, and Gunnison East contains a heterogeneous mix of bare ground, deciduous forest, and wooded grassland. These differences in land cover affect snow accumulation and ablation processes. In forested areas (Aspen and Gunnison-East), canopy interception and sublimation reduce the fraction of snowfall that reaches the ground, while canopy shading affects the surface energy balance and generally slows melt.

The three forcing datasets have similar relative snowfall distributions across watersheds and elevation bands (with MERRA2 yielding the largest snowfall, followed by NLDAS2 and ERA5, which are relatively comparable as shown in Fig. 2). However, the ASO SWE exhibits more significant differences between watersheds, leading to the varying performance of the forcing datasets. For example, ASO peak SWE in Merced is higher than peak SWE in Aspen and Gunnison East which also exhibit higher SWE later into the season. These differences across watersheds are likely in part due to differences in forest cover, which leads to more canopy interception and slower melt rates in Aspen and Gunnison-East. This indicates that differences in SWE performance are not driven only by snowfall magnitude, but by how snowfall errors distribute and propagate nonlinearly into SWE under specific land cover conditions.

(2) Elevation and downscaling of forcing datasets: ERA5, MERRA2, and NLDAS2 are different in their raw resolutions and associated grid mean elevations. As described in Sect. 2.3 and Supplement S1, all meteorological variables are downscaled to the 150 m model grid by explicitly accounting for elevation differences between the coarse forcing grids and the high-resolution SRTM DEM used in the land surface model. For each dataset, forcing variables are first spatially interpolated, and the elevation associated with the forcings is obtained through interpolation of the raw elevations. The interpolated forcings are then topographically corrected and projected onto the SRTM elevations at 150 m resolution.

During downscaling, meteorological variables (e.g., air temperature) are corrected using lapse rate corrections that explicitly depend on the elevation difference ($\Delta Z$) between the forcing dataset and the SRTM DEM. Therefore, discrepancies between the raw forcing elevations and SRTM elevations propagate directly into the downscaled forcings. Because basins differ in their elevation distributions, as represented by the SRTM DEM, the same forcing dataset can perform differently across watersheds, including adjacent

basins such as Aspen and Gunnison-East that share the same native MERRA2 grid cell (Fig. 1). As shown in Fig. S2, despite identical raw MERRA2 precipitation forcing, Aspen exhibits systematically more positive $\Delta Z$ values (with more spread) than Gunnison-East. Air temperature is spatially distributed using a fixed lapse rate applied to the elevation difference $\Delta Z$. Consequently, larger elevation differences lead to larger temperature adjustments, which directly influence the rain-snow partitioning. As Aspen exhibits larger positive $\Delta Z$ than Gunnison-East, it has systematically lower corrected temperatures and therefore higher snowfall and peak SWE than Gunnison-East.

As discussed in the last paragraph of Sect. 2.3, topographic adjustment and bias correction cannot fully remove representation errors inherent to coarse scale forcing datasets. Large scale precipitation, air temperature, and radiation still smooth over complex mountainous terrain, and product-dependent biases still exist after downscaling. These contribute to the inter-dataset spread in annual snowfall and propagate to differences in prior SWE. Therefore, differences in SWE performance primarily reflect a combination of raw forcing errors and elevation representation errors relative to the SRTM DEM.

To address the reviewer's comment, we propose the following revisions:

1. Add land cover maps as a new supplementary figure (Fig. S1) and describe land cover in Sect. 2.1. Add a new paragraph at the end of Sect. 2.1. "The three watersheds also exhibit different dominant land cover types that influence snow accumulation and ablation processes. Land cover type and fractional forest cover are derived from the AVHRR land cover and GLCF forest cover datasets and interpolated to the 150 m model grid using a nearest neighbor interpolation. As shown in Fig. S1, Merced is dominated by grassland and wooded grassland, with relatively limited dense forest cover. Aspen exhibits a transition from deciduous forest at lower elevations to wooded grassland and alpine terrain at higher elevations. Gunnison-East contains a heterogeneous mix of bare ground, deciduous forest, and wooded grassland. These differences influence canopy interception, sublimation, and radiative fluxes."

2. Add an interpretation of prior SWE differences based on land cover type. In line 316 after "negative differences relative to the ASO-based reference in April", add "Beyond differences among forcing datasets, the ASO-based reference itself exhibits spatial variability across the three watersheds. As shown in Fig. 4, ASO peak SWE in Merced is higher than peak SWE in Aspen and Gunnison East which also exhibit higher SWE later into the season. These differences across watersheds are likely in part due to differences in forest cover, which leads to more canopy interception and slower melt rates in Aspen and Gunnison-East."

3. Explicitly explain elevation handling and reference DEM. At the end of the first paragraph in Sect.2.3, add "All meteorological variables are downscaled to the 150 m model grid by explicitly accounting for elevation differences between the coarse forcing grids and the high-resolution SRTM DEM used in the land surface model. Discrepancies between raw forcing elevations and SRTM elevations propagate directly into the downscaled forcings."

4. Explicitly introduce elevation-dependent differences in snowfall that are caused by differences in raw elevation relative to SRTM DEM and illustrate this effect with a new supplementary figure (Fig. S2). Replace the sentence in line 182-183 "Differences among datasets …" with "Differences among datasets are consistent across many elevation bands due to the coarse resolution of the raw forcing products and the use of the same deterministic downscaling approach. In addition, part of the elevation-dependent snowfall differences arises from differences between the native elevations of the coarse forcing datasets and the high-resolution SRTM DEM used for downscaling. ERA5, MERRA2, and NLDAS2 are different in spatial resolution and associated elevations, and air temperature is adjusted based on the elevation difference ($\Delta Z$) between the SRTM DEM and interpolated forcing DEM; therefore, any errors in the applied lapse rates scale with basin-specific elevation distributions. This effect is illustrated in Fig. S2, which shows that although Aspen and Gunnison-East share the same native MERRA2 grid cell (Fig. 1), Aspen exhibits systematically more positive $\Delta Z$ (with more spread) in SRTM DEM than Gunnison-East. Air temperature is spatially distributed using a fixed lapse rate applied to the elevation difference $\Delta Z$. Consequently, larger elevation differences lead to larger temperature adjustments, which directly influence the rain-snow partitioning. As Aspen exhibits larger positive $\Delta Z$ than Gunnison-East, it has systematically lower corrected temperatures and therefore higher snowfall and peak SWE than Gunnison-East. As a result, adjacent basins with different elevation distributions can experience different snowfall partitioning even when forced by the same dataset."

5. Clarify the inter-product spread. Replace the sentence in line 189-190 "The observed inter-product variability reflects …" with "The observed inter-product variability reflects a combination of inherent differences in the raw datasets, elevation-dependent errors from the downscaling of coarse-scale forcings over complex terrain, and the spatially varying impacts of the applied bias corrections."

[Figure]

Figure S1. Land cover type (left) and fractional forest cover (right) for the Merced, Aspen, and Gunnison-East watersheds at 150 m resolution.

[Figure]

Figure. S2: (a) Spatial map of the elevation difference ΔZ (SRTM − MERRA2 interpolation) over the Aspen and Gunnison-East watersheds used in the downscaling of non-precipitation forcing variables. (b) Violin plots showing the distributions of ΔZ for each watershed. The black horizontal line denotes the median elevation, with the interquartile range (IQR) indicated by the vertical black bars.

**MINOR COMMENTS**

1. Figure 1: Would be interesting to add the land cover map, which is used in resampling (Figure 3).

Response: We propose to add the land cover map (Fig. S1) as a new supplementary figure, and reference it in the main text, because Fig.1 already contains multiple panels and is visually dense.

2. Eqns. 1-3: Personally, I prefer the more compact ways of writing these equations. They could shorten into one equation $W_i = \frac{\prod_{j \neq i} \sigma_j^2}{\sum_i \prod_{j \neq i} \sigma_j^2}$

Response: Suggestion adopted. We will use the compact equation:

$$W_i = \frac{\prod_{j \neq i} \sigma_j^2}{\sum_i \prod_{j \neq i} \sigma_j^2}$$

3. Eqns. 4-6: Same as the previous equations, it could be summarized by $N_i = W_i \times N$

Response: Suggestion adopted. We will use the compact equation:

$$N_i = W_i \times N$$

where $N = \sum_i N_i$

4. L. 321: Please clarify what is meant by "SWE variations". I assume it is the difference with ASO SWE observations.

Response: "SWE variations" is meant to represent the maximum difference among the three prior SWE estimates. We've revised the sentence to: "On the near-peak ASO date, the SWE variation, defined as the maximum difference among the three prior SWE estimates, is equal to 0.46 m in Merced, 0.64 m in Aspen, and 0.51 m in Gunnison-East.".

5. Figure 4: it would be interesting to add error bars on the ASO measurements to show the spatial variability of SD and SWE across the different study areas and the spread in SWE due to the mean ensemble density estimation. You show which forcing data works better on average, but the error bars could indicate whether the different forcing datasets provide estimates within the range of measured properties.

Response: We appreciate the reviewer's suggestion to include error bars on the ASO SD and SWE estimates in Fig.4. We agree that uncertainty information is important when interpreting model performance relative to observations. However, we chose not to include error bars in this figure for two reasons: (1) the uncertainty associated with ASO measurements in our framework represents measurement error used within the data assimilation, instead of spatial variability of SWE or SD across the watershed. The measurement error is not intended to quantify basin scale variability or uncertainty in basin-averaged SWE and SD. Including these errors as bars on basin-averaged time series could be misleading. (2) Fig. 4 presents daily time series of basin-averaged SWE and SD, whereas spatial variability is analyzed in elevational distribution and spatial SWE differences. (3) ASO SWE uncertainty due to snow density is relatively small (Fig. R1). Basin-averaged SWE estimates adjusted using snow density derived from ERA5, MERRA2, NLDAS2, and their mean show minimal spread and is therefore deemed a small contributor to inter-product uncertainty.

[Figure]

Figure R1. Basin-average ASO SWE adjusted by prior snow density from ERA5, MERRA2, NLDAS2, and their average.

6. Table 3: Probably a precision error but weights do not add up to 1 for Merced.

Response: Thanks for the comment. The number of replicates sum up to the total number of replicates. The existing weights do not add up to 1 due to a precision error. We propose to revise the weights using more significant digits for Merced in Table 3.

**Table 3.** Partitioning weights ($W$) and number of realizations used in the multi-forcing ensemble for each domain based on the RMSE of prior mean SWE relative to ASO-based SWE near April $1^{st}$.

| Domains | | ERA5 | MERRA2 | NLDAS2 |
|---|---|---|---|---|
| Merced | $W$ | 0.55 | 0.125 | 0.325 |
| | # of realizations | 66 | 15 | 39 |
| Aspen | $W$ | 0.34 | 0.17 | 0.49 |
| | # of realizations | 41 | 20 | 59 |
| Gunnison-East | $W$ | 0.33 | 0.38 | 0.29 |
| | # of realizations | 39 | 46 | 35 |

7. Figure 7: I understand the idea behind keeping only the best and worst cases in this figure but I think it would still be relevant to include all three forcing cases. I would also keep the same order of presentation in the figure as previous figures to make it easier to compare. I also like to keep the y-axis range the same when comparing plots horizontally. It helps to identify which site has larger errors more easily.

Response: We appreciate the reviewer's suggestion. We propose to revise Fig. 7 and Fig. 9 (as shown below) to include all forcing cases in the same order of presentation, and keep the y-axis range the same.

[Figure]

Figure 7. Bar plots of the RMSE, absolute bias, and ubRMSE of prior mean SWE compared to ASO-based SWE across all domains and ASO observation times. The rightmost column summarizes the overall statistics across all domains and dates.

[Figure]

Figure 9. Same as Fig.7, but for the posterior mean SWE. The added light grey shaded area highlights the assimilation date.

**Reply to Reviewer #2 Comments**

**MAJOR COMMENTS**

The most major critique I have of the paper is that it does not include a sampling of hydrologic conditions (i.e., wet, dry, and average years). The period analyzed (WY 2019) was a wet, high snow accumulation year in both the California Sierra Nevada and the Colorado Rocky Mountains. The main justification for this year seems to be that lidar snow data were available across all three basins (L. 111-112), but I don't think it is necessary that the same year be used in distinct regions (CA vs. CO). It is the choice of the authors whether or not to bring additional years (e.g. dry, average) into the analysis, but at a minimum I think the paper should provide more description of the snow/weather conditions in the study year(s) and include some discussion on how the type of snow year may influence the DA (e.g., see Margulis et al. 2019, GRL).

Response: We thank the reviewer for this important comment regarding the representativeness of hydrologic conditions and the influence of the type of snow year on DA performance. We agree that WY2019 was a relatively wet year in both Sierra Nevada and Colorado Rocky Mountains. This choice of WY 2019 was driven primarily by the availability of coincident, multiple-dates ASO lidar data across all three watersheds. In particular, WY 2019 is the year for which ASO data were available both near peak accumulation and during the melt season. Focusing on a wet year also provides a meaningful test case for DA performance because longer melt periods allow errors in meteorological forcing to propagate, thereby increasing the sensitivity of SWE estimates to forcing uncertainty. Limiting the analysis to a single wet year does not constrain the generality of the conclusions that integrating diverse meteorological forcing within a DA framework improves SWE estimates, especially where the optimal forcing dataset is uncertain. In response to the suggestion, we propose to explicitly describe the snow conditions for WY 2019 and add discussion on how snow year type may influence DA performance. While extending the analysis to multiple years would be valuable, doing so requires additional ASO coverage that is not currently available across all three watersheds. We therefore frame this study as a focused demonstration under data rich, high snow conditions, and clarify multi-year analysis as an important potential direction for future work.

To address the reviewer's comment, we propose the following revisions:

1. Add a sentence in line 111-112 to describe the snow conditions "WY 2019 was characterized by well above average snow accumulation across much of the western United States, making it a wet snow year relative to long-term climatology. In California, peak statewide snowpack was 175 percent of average with records dating back to 1950 according to the hydroclimate report from California Department of Water Resources."

2. Clarify scope and limitations of using a single wet year. Add a sentence to the end of the first paragraph of Sect. 2.1 "We acknowledge that restricting the analysis to a single WY limits the hydroclimate conditions examined. Therefore, this study is a demonstration of a

representative wet snow year. Extending the analysis across multiple snow regimes would be valuable but is currently limited by the availability of consistent ASO observations across all study domains."

Add a new sentence at the end of the conclusions "Future work should extend this framework to include dry, average, and wet snow years to characterize how hydroclimate region influences forcing sensitivity and data assimilation performance. "

3. Add a discussion regarding the impact of snow conditions on DA performance. Add the paragraph at the end of the first paragraph in Sect. 3.3 "Previous work by Margulis et al. (2019) demonstrated that in wet years with deep snowpack, assimilation near peak accumulation tends to be most effective. In contrast, in an extreme record-dry year (WY 2015), DA may be less effective if prior ensembles contain limited snow or if observations occur after partial melt, leading to weaker updates. The significant DA impacts observed in this study are therefore consistent with the wet-year case analyzed by Margulis et al. (2019). Extension of this analysis to dry and normal snow years would be valuable for assessing the DA performance across hydroclimate regimes and is a key direction for future work."

**GENERAL COMMENT:**

I find that Section 3 is more of a "Results" section than a "Results and Discussion" section (as intended). I see minimal elements that make a classic discussion section – e.g., comparisons to other studies, discussions of future research needs, etc. I would suggest adding more discussion elements throughout section 3 (as appropriate) or alternatively making a short subsection at the end of section 3 that provides a more substantive discussion.

Response: We appreciate the reviewer's comment. In response, we propose to revise Sect. 3 to strengthen its discussion within subsections.

1. Sect. 3.1: Add an interpretation of prior SWE differences based on land cover type. In line 316 after "negative differences relative to the ASO-based reference in April", add "Beyond differences among forcing datasets, the ASO-based reference itself exhibits spatial variability across the three watersheds. As shown in Fig. 4, ASO peak SWE in Merced is higher than peak SWE in Aspen and Gunnison East which also exhibit higher SWE later into the season. These differences across watersheds are likely in part due to differences in forest cover, which leads to more canopy interception and slower melt rates in Aspen and Gunnison-East."

2. Sect. 3.1: Add a discussion of the impact of elevation on forcing disaggregation and prior SWE in line 358. "ERA5 and NLDAS2 exhibit similar RMSE in the range of 0.3-0.45 m in Aspen and Gunnison-East. However, a counterintuitive result is that MERRA2 exhibits different weights in Aspen and Gunnison-East, despite being adjacent watersheds and falling within the same native MERRA2 grid cell. While raw MERRA2 precipitation is identical for both basins

prior to downscaling, the snowfall forcing is different due to elevation-dependent temperature corrections and basin-specific elevation distributions. As shown in Fig. S2a, bilinearly interpolated MERRA2 elevations differ from SRTM elevations in both magnitude and spatial pattern. Aspen has a higher median elevation and a larger elevation range than Gunnison-East as represented by SRTM (Fig. S2b). These differences directly influence the lapse rate-based temperature correction applied during forcing disaggregation. Therefore, Aspen experiences lower corrected temperatures, leading to a higher fraction of precipitation falling as snow and greater peak SWE, whereas Gunnison-East exhibits warmer corrected temperatures and reduced snowfall." And add a sentence to line 365 "In contrast, MERRA2 exhibits lower RMSE in Gunnison-East on April 7th, 2019, where random errors dominate and its elevation adjusted snowfall aligns better with ASO-based SWE."

3. As done in Major comments: Add a discussion regarding the impact of snow conditions on DA performance. Add the paragraph at the end of the first paragraph in Sect. 3.3 "Previous work by Margulis et al. (2019) demonstrated that in wet years with deep snowpack, assimilation near peak accumulation tends to be most effective. In contrast, during dry years, DA may be less effective if prior ensembles contain limited snow or if observations occur after partial melt, leading to weaker updates. The significant DA impacts observed in this study are therefore consistent with the wet-year case analyzed by Margulis et al. (2019). Extension of this analysis to dry and normal snow years would be valuable for assessing the DA performance across hydroclimate regimes and is a key direction for future work."

A result that is interesting but not discussed in detail is that there are quite different weights for Aspen versus Gunnison-East (e.g., Tables 3 and 5). This is surprising (at least to me), considering that they are adjacent basins (Fig. 1). Why is this result obtained and what might it suggest about the forcing data and/or the snow in these basins?

Response: We thank the reviewer for highlighting this point. Although Aspen and Gunnison-East are geographically adjacent and fall within the same MERRA2 grid cell (Figure 1), differences in forcing weights can arise from elevation-driven differences in snowfall after downscaling. To smooth the coarse-resolution MERRA2 and reduce grid-scale artifacts, the raw forcing (e.g., MERRA2) is first interpolated to the model grid using bilinear interpolation. As shown in Fig. S2 (see response to Reviewer 1), the interpolated elevation from MERRA2 exhibits differences from SRTM elevation, and these elevation differences ($\Delta Z$) vary spatially between the two basins. The violin plots in Fig. S2 shows that Aspen has a higher median $\Delta Z$ and larger elevation difference distribution than Gunnison-East, indicating a larger discrepancy between MERRA2 elevation and the SRTM DEM in Aspen.

These elevation differences directly affect the temperature lapse rate correction applied during forcing disaggregation. The forcing weights are derived by comparing modeled near-peak SWE to ASO observations, which are primarily controlled by accumulation-season snowfall. Because total precipitation from MERRA2 is nearly identical for the two basins (as they share the same

MERRA2 grid cell), any differences in snowfall most likely arise from differences in air temperature disaggregation. Air temperature is spatially distributed using a fixed lapse rate applied to the elevation difference between the native forcing DEM (MERRA2) and the SRTM DEM. Consequently, larger elevation differences lead to larger temperature adjustments, which directly influence the rain-snow partitioning. Any errors in the fixed lapse rate will be enhanced by larger elevation differences ($\Delta Z$).

As Aspen exhibits systematically larger $\Delta Z$ (and more spread) than Gunnison-East, it experiences larger temperature corrections, resulting in lower corrected temperatures, increased snowfall, and higher peak SWE. Overall, the different forcing weights result from a combination of raw forcing elevation errors and the effects of disaggregation. The weighting framework used in this paper is based on the RMSE between modeled SWE and the ASO SWE reference, not on geographic proximity. Each basin preferentially weights the forcing dataset whose elevation adjusted snowfall most closely matches the observed SWE. Therefore, even adjacent basins can yield different forcing weights when their elevation distributions and temperature corrections differ.

**Line Comments**

L. 14-20: One nuance that is not conveyed clearly here is that the multi-forcing reduces errors relative to most forcing datasets, but not all forcing datasets. As written, it sounds like the multi-forcing is always the most accurate. Can you convey this nuance while also indicating that the "best" forcing dataset cannot be known a priori, and the "best" dataset may vary in space and time?

Response: Suggestion adopted. We propose to revise the paragraph: The multi-forcing ensemble generally reduces errors compared to most individual forcing datasets and improves prior SWE accuracy across the study regions. Assimilation of near-peak lidar-derived snow depth substantially corrects prior SWE errors, reducing the influence of forcing-driven biases accumulated during the snowfall season. As a result, random error is the dominant source of posterior error. Although assimilation narrows performance differences, the multi-forcing ensemble still yields slightly better overall accuracy and improved uncertainty characterization. This work demonstrates that integrating diverse meteorological forcings within a data assimilation framework can improve SWE estimates (both model-based and reanalysis-based) when the optimal forcing dataset cannot be identified a priori and varies across space and time.

L. 45: "transboundary" is often used in water studies in regard to rivers that cross international political boundaries, which is not true for all the mountain ranges referenced here (e.g., Sierra Nevada). Please reword.

Response: Suggestion adopted. We propose to revise the sentence: Coarse resolution products often fail to capture snow storage patterns in  rain shadow mountain regions (e.g., Sierra Nevada and Andes) where snowmelt feeds watersheds that supply distinct downstream populations (Fang et al., 2023).

L. 47: To be more exact, I suggest replacing "estimates" with "process-based estimates" or "hydrological model estimates". Physical versus statistical approaches for estimating runoff are impacted differently by snow data uncertainty, and I think the sentence is more relevant to the former.

Response: Suggestion adopted. We propose to replace estimates of runoff with process-based estimates of runoff.

L. 91: What does "readily available" mean in this context?

Response: By "readily available", we mean widely used, publicly accessible meteorological forcing datasets, specifically ERA5, MERRA2, and NLDAS2. We propose to revise the sentence: Does one of the  widely used meteorological forcing datasets (ERA5, MERRA2, or NLDAS2) yield the most accurate model-based prior SWE spatio-temporal estimates?

L. 125-126: Need to add downwelling longwave radiation here?

Response: Suggestion adopted. We propose to revise the sentence to "… surface downwelling shortwave and longwave radiation, …"

L. 167-168: Broxton et al. (2016) may also be relevant here.

Response: Suggestion adopted. We propose to add the citation to "… datasets, and evaluations of global reanalysis products have identified systematic underestimation of SWE associated with forcing and model representation errors (Broxton et al., 2016), suggesting that similar biases are expected in ERA5. "

L. 270: Suggest include the RMSE^2 equation as a distinct/numbered equation (#4).

Response: Suggestion adopted. We propose to revise the sentence to "The RMSE can be decomposed into bias and unbiased RMSE (ubRMSE) components according to Entekhabi et al. (2010):

$$RMSE^2 = bias^2 + ubRMSE^2 \quad (Eq. 4)"$$

L. 286: For clarity, make this "(i.e., N=120)".

Response: Suggestion adopted. We propose to replace the "(i.e., 120)" to "(i.e., N=120)".

L. 321: Remove "variations".

Response: "SWE variations" is meant to represent the maximum difference among the three prior SWE estimates. We propose to revise the sentence to "On the near-peak ASO date, the SWE variation, defined as the maximum difference among the three prior SWE estimates, equals 0.46 m in Merced, 0.64 m in Aspen, and 0.51 m in Gunnison-East".

L. 322: Replace "differences" with "ranges"?

Response: Same as L. 321, We propose to replace "Corresponding SD differences" with "Corresponding SD variations"

L. 342-345 and Fig. 5: The SWE depth versus elevation plots are useful, but I would argue that not all elevation bands are "equal" in a hydrologic sense and a snow storage sense, considering that there may be very different amounts of land area contained in each elevation band (depending on the hypsometry). It would be useful to know how this looks for SWE volume versus elevation, perhaps as a supplementary figure?

Response: We thank the reviewer for this insightful comment. Figure 5 was designed to show elevation dependent differences in mean SWE depth, which is useful for diagnosing how forcing performance varies with elevation, but it does not directly represent the contribution of each elevation band to total basin snow storage. That said, the elevation bins were not uniform but chosen to include the same number of pixels, which should lead to each bin having similar land areas. To confirm this, we have added Fig. R3 that shows the distribution of SWE volume as a function of elevation, computed by integrating SWE over the land area within each elevation band. Because the elevation bins were constructed to contain equal numbers of pixels, each bin represents approximately the same land area within the watershed. As a result, SWE volume within each elevation bin is proportional to mean SWE, yielding similar elevational patterns for SWE depth.

[Figure]

Figure R3. Elevational distribution of ASO-based SWE volume and prior mean SWE volume produced using ERA5, MERRA2, and NLDAS2 forcings across (a) Merced, (b) Aspen, and (c) Gunnison-East.

L. 400-401: Note here that the absolute bias of the multi-forcing is still higher than ERA5 and NLDAS2.

Response: We propose to revise the sentence to: For absolute bias, the multi-forcing case significantly reduces errors to 0.24 m, compared to 0.49 m for MERRA2, but still higher than ERA5 and NLDAS2.

L. 444-446: I do not disagree with this discussion point. However, I think it may also be worth noting that there are quite different time periods for the accumulation season vs. the ablation season, and that means there is more opportunity for errors (bias) to build up in the accumulation season. The accumulation season may be two to three times longer in duration than the ablation season, and in this study the "ablation season" is only partial because the ASO survey occur part of the way through the ablation season (i.e., before complete melt out).

Response: Suggestion adopted. We propose to revise the sentence to " This comparison indicates that winter accumulation forcings likely contribute on the order of twice as much SWE error as melt-season forcings, making them the dominant source of uncertainty in the model. Note that

the accumulation season RMSE is based on peak SWE near the end of the accumulation period, whereas the melt-season RMSE is derived from mid-melt season and corresponds only to a partial ablation period prior to complete melt out."

L. 471: Add "(Fig. 7)" after "unknown".

Response: Suggestion adopted.

L. 480-481: This is a great point and one that the community should appreciate on the value of SD data assimilation.

Response: We agree.

L. 527-528: Add "(class 0)" after "ASO-based SWE" and "(class 3)" after "none of them do" to help clarify the conventions.

Response: Suggestion adopted.

**Figures and Tables**

Figure 1: In the lower right panel, suggest rounding the mean value to the nearest mm.

Response: Suggestion adopted.

[Figure]

Figure 7 and Figure 9: I found these confusing and it took me some time to finally figure out what they are showing. At first I thought that some of the forcing cases were missing, and it wasn't clear to me until I read the results text that the "middle" case (not best, not worst) was being omitted. For clarity, consistency, and completeness, I think it would make sense to include all 3 forcing scenarios and the multi-forcing (as in the right panels). You could denote the best/worst by placing a marker above the corresponding bar.

Response: Suggestion adopted. Same as the response to reviewer 1 (Minor Comment 7).

Figure 12: Would this be better displayed as a table rather than a figure?

Response: We thank the reviewer for this suggestion. We agree that the information shown in Fig. 12 could be presented in a table. However, we chose to keep Fig. 12 to facilitate the visual comparison across forcing datasets, basins, and experimental configurations.

References

Broxton, P. D., Zeng, X., and Dawson, N.: Why Do Global Reanalyses and Land Data Assimilation Products Underestimate Snow Water Equivalent?, Journal of Hydrometeorology, 17, 2743–2761, https://doi.org/10.1175/JHM-D-16-0056.1, 2016.

Margulis, S. A., Fang, Y., Li, D., Lettenmaier, D. P., and Andreadis, K.: The Utility of Infrequent Snow Depth Images for Deriving Continuous Space-Time Estimates of Seasonal Snow Water Equivalent, Geophysical Research Letters, 46, 5331–5340, https://doi.org/10.1029/2019GL082507, 2019.